# IDENTIFIABILITY OF LABEL NOISE TRANSITION MATRIX

## ABSTRACT

The noise transition matrix plays a central role in the problem of learning with noisy labels. Among many other reasons, a large number of existing solutions rely on access to it. Identifying and estimating the transition matrix without ground truth labels is a critical and challenging task. When label noise transition depends on each instance, the problem of identifying the instance-dependent noise transition matrix becomes substantially more challenging. Despite recent works proposing solutions for learning from instance-dependent noisy labels, the field lacks a unified understanding of when such a problem remains identifiable. The goal of this paper is to characterize the identifiability of the label noise transition matrix. Building on Kruskal's identifiability results, we show the necessity of multiple noisy labels in identifying the noise transition matrix for the generic case at the instance level. We further instantiate the results to relate to the successes of the state-of-the-art solutions and how additional assumptions alleviated the requirement of multiple noisy labels. Our result also reveals that disentangled features are helpful in the above identification task and we provide empirical evidence.

## 1 INTRODUCTION

The literature of learning with noisy labels concerns the scenario when the observed labels $\tilde{Y}$ can differ from the true one $Y$. The noise transition matrix $T(X)$, defined as the transition probability from $Y$ to $\tilde{Y}$ given $X$, plays a central role in this problem. Among many other benefits, the knowledge of $T(X)$ has demonstrated its use in performing either risk (Natarajan et al., 2013; Patrini et al., 2017a), or label (Patrini et al., 2017a), or constraint corrections (Wang et al., 2021a). In beyond, it also finds applications in ranking small loss samples (Han et al., 2020) and detecting corrupted samples (Zhu et al., 2021a). On the other hand, applying the wrong transition matrix $T(X)$ can lead to a number of issues. The literature has well-documented evidence that a wrongly inferred transition matrix can lead to performance drops (Natarajan et al., 2013; Liu & Wang; Xia et al., 2019; Zhu et al., 2021c), and false sense of fairness (Wang et al., 2021a; Liu & Wang). Knowing whether a $T(X)$ is identifiable or not helps understand if the underlying noisy learning problem is indeed learnable. Prior works have documented challenges in estimating the noise transition matrices when the quality of available training information remains unclear. For instance, in (Zhu et al., 2022) the authors show that when the quality of representations dropped, the estimation error in $T(X)$ increases significantly (Figure 1 therein). Other previous references have documented these challenges too (Xia et al., 2019). We have also provided experiments to validate the argument in Appendix C.4.

The earlier results have focused on class- but not instance-dependent transition matrix $T(X) \equiv T := [\mathbb{P}(\tilde{Y} = j | Y = i)]_{i,j}, \forall X$. The literature has provided discussions of the identifiability of $T$ under the mixture proportion estimation setup (Scott, 2015), and has identified a reducibility condition for inferring the inverse noise rate. Later works have developed a sequence of solutions to estimate $T$ under a variety of assumptions, including irreducibility (Scott, 2015), anchor points (Liu & Tao, 2016; Xia et al., 2019; Yao et al., 2020a), separability (Cheng et al., 2020), rankability (Northcutt et al., 2017; 2021), redundant labels/tensor (Liu et al., 2020; Traganitis et al., 2018; Zhang et al., 2014), clusterability (Zhu et al., 2021c), among others (Zhang et al., 2021; Li et al., 2021).

Recent study (Wei et al., 2021) has empirically shown that the above class-dependent model is not precise in capturing the real-world noise patterns, but rather real human-level noise follows an instance-dependent model. Intuitively, the instance $X$ encodes the difficulties in generating the label

for it. This more realistic, flexible and powerful noise model helps characterize the challenges. We observe a recent surge of different solutions towards solving the instance-dependent label noise problem (Cheng et al., 2020; Xia et al., 2020b; Cheng et al., 2021a; Yao et al., 2021). Some of the results took on the problem of estimating $T(X)$, while the others proposed solutions to learn directly from instance-dependent noisy labels. We will survey these results in Section 1.1. The question of identifying and estimating $T$ becomes much trickier when the noise transition matrix is instance-dependent. The potentially complicated dependency between $X$ and $T(X)$ renders it even less clear whether solving this problem is viable or not.

Despite the above successes, there lacks a unified understanding of when this learning from instance-dependent noisy label problem is indeed identifiable and therefore learnable. The mixture of different observations calls for the need for demystifying: (1) Under what conditions are the noise transition matrices $T(X)$ identifiable? (2) When and why do the existing solutions work when handling the instance-dependent label noise? (3) When $T(X)$ is not identifiable, what can we do to improve its identifiability? Providing answers to these questions will be the primary focus of this paper. The main contributions of this paper are to characterize the identifiability of instance-dependent label noise, use them to provide evidences to the success of existing solutions and point out possible directions to improve. Among other findings, some highlights of the paper are 1. We find many existing solutions have a deep connection to the celebrated Kruskal's identifiability results that date back to the 1970s (Kruskal, 1976; 1977). 2. Three separate independent and identically distributed (i.i.d.) noisy labels (random variables) are both necessary and sufficient for instance-level identifiability. This observation echoes the previous successes of developing tensor-based approaches for identifying the hidden models. 3. Disentangled features help with identifiability. Our paper will proceed as follows. Section 2 and 3 will present our formulation and the highly relevant preliminaries. Section 4 provides characterizations of the identifiability at the instance level and lays the foundations for our discussions. Section 5 extends the discussion to different instantiations that help us provide evidences to the success of existing solutions. Section 6 provides some empirical observations.

## 1.1 RELATED WORKS

In the literature of learning with label noise, a major set of works focus on designing *risk-consistent* methods, i.e., performing empirical risk minimization (ERM) with specially designed loss functions on noisy distributions leads to the same minimizer as if performing ERM over the corresponding unobservable clean distribution. The *noise transition matrix* is a crucial component for implementing risk-consistent methods, e.g., loss correction (Patrini et al., 2017b), loss reweighting (Liu & Tao, 2015), label correction (Xiao et al., 2015) and unbiased loss (Natarajan et al., 2013). A number of solutions were proposed to estimate this transition matrix for class-dependent label noise, which we have discussed in the introduction. To handle instance-dependent noise, recent solutions include estimating local transition matrices for different groups of data (Xia et al., 2020b), using confidence scores to revise transition matrices (Berthon et al., 2020), and using clusterability of the data (Zhu et al., 2021c). More recent works have used the causal knowledge to improve the estimation (Yao et al., 2021), and the deep neural network to estimate the transition matrix defined between the noisy label and the Bayes optimal label (Yang et al., 2021). Other works chose to focus on the learning from instance-dependent label noise directly, without explicitly estimating the transition matrix (Zhu et al., 2021b; Cheng et al., 2021a; Berthon et al., 2021; Xia et al., 2020a; Li et al., 2020).

The identifiability issue with label noise has been discussed in the literature, despite not being formally treated. Relevant to us is the identifiability results studied in the Mixture Proportion Estimation setting (Scott, 2015; Yao et al., 2020b; Menon et al., 2015). We'd like to note that the identifiability was defined for the inverse noise rate, which differs from our focus on the noise transition matrix $T$. To our best knowledge, we are not aware of other works that specifically address the identifiability of $T(X)$, particularly for an instance-dependent label noise setting. Highly relevant to us is the Kruskal's identifiability results (Kruskal, 1976; 1977; Sidiropoulos & Bro, 2000; Allman et al., 2009), which reveals a sufficient condition for identifying a parametric model that links a hidden variable to a set of observed ones. Kruskal's early results were developed under the context of tensor, which later proves to be a powerful tool for learning latent variable models (Sidiropoulos et al., 2017; Zhang et al., 2014; Anandkumar et al., 2014).

## 2 FORMULATION

We use $(X, Y)$ to denote a supervised data in the form of (feature, label) drawn from an unknown distribution over $X \times Y$. We consider a $K$-class classification problem where the label $Y \in \{1, 2, ..., K\}$ with $K \geq 2$. In our setup, we do not observe the clean true label $Y$, but rather a noisy one, denoting by $\tilde{Y}$. The generation of $\tilde{Y}$ follows the following transition matrix: $T(X) := [\mathbb{P}(\tilde{Y} = j | Y = i, X)]_{i,j=1}^{K}$. $T(X)$ is a $K \times K$ matrix with its $(i, j)$ entry being $\mathbb{P}(\tilde{Y} = j | Y = i, X)$.

To define identifiability, we will denote by $\Omega$ an observation space. We first define identifiability for a general parametric space $\Theta$. Denote the distribution induced by the parameter $\theta \in \Theta$ of a statistical model on the observation space $\Omega$ as $\mathbb{P}_{\theta}$ (Kruskal, 1976; Allman et al., 2009). To give an example, for a fixed $X$ (when consider instance-level identifiability), and $\Omega$ is simply the outcome space for its associated noisy label $\tilde{Y}$, i.e., $\{1, 2, ..., K\}$. In this case, each $\theta$ is the combination of a possible transition matrix $T(X)$ and the hidden prior of $\mathbb{P}(Y|X)$, which we use to denote the conditional probability distribution of $Y$ given $X$. $\mathbb{P}_{\theta}$ is then the distribution (probability density function) $\mathbb{P}(\tilde{Y}|X)$. Later in Section 4 when we introduce three noisy labels $\tilde{Y}_1, \tilde{Y}_2, \tilde{Y}_3$ for each $X$, $\mathbb{P}_{\theta}$ is the joint distribution $\mathbb{P}(\tilde{Y}_1, \tilde{Y}_2, \tilde{Y}_3 | X)$. Identifiability defines as follows:

**Definition 1** (Identifiability). *The parameter $\theta$ (statistical model) is identifiable if $\mathbb{P}_{\theta} \neq \mathbb{P}_{\theta'}, \forall \theta \neq \theta'$.*

We define identifiability for the task of learning with noisy labels for an $X$. Denote by $\theta(X) := \{T(X), \mathbb{P}(Y|X)\}$. $\mathbb{P}_{\theta(X)}$ is the distribution (probability density function) over $\Omega$, defined by the noise transition matrix $T(X)$ and the prior $\mathbb{P}(Y|X)$. To emphasize, $\Omega$ is not necessarily the observation space of the noisy label $\tilde{Y}$ only. The exploration of an effective $\Omega$ will be one of the focuses.

**Definition 2** (Identifiability of $T(X)$). *For a given $X$, $T(X)$ is identifiable if $\mathbb{P}_{\theta(X)} \neq \mathbb{P}_{\theta'(X)}$ for $\theta(X) \neq \theta'(X)$, up to label permutation.*

Label permutation relabels the label space, e.g., $1 \rightarrow 2, \ 2 \rightarrow 1$, and the rows in $T(X)$ will swap. Allowing for label permutation would mean that our results allow the high noise rate regime. For instance, for a binary classification problem, an 80% noise rate would correspond to a counterfactual 20% one. Finding either model would be regarded as being identifiable. In practice, further restriction such as noise rate should not exceed 50% can help us remove one of the two cases.

## 3 PRELIMINARY

In this section, we will introduce two highly relevant results on Mixture Proportion Estimation (MPE) (Scott, 2015) and Kruskal's identifiability result (Kruskal, 1976; 1977).

### 3.1 PRELIMINARY RESULTS USING IRREDUCIBILITY AND ANCHOR POINTS

The problem of learning from noisy labels ties closely to another problem called Mixture Proportion Estimation (MPE) (Scott, 2015), which concerns the following problem: let $F, J, H$ be distributions defined over a Hilbert space $\mathcal{Z}$. The three relate to each other as follows: $F = (1 - \kappa^*)J + \kappa^* H$. The identifiability problem concerns the ability to identify the mixture proportion $\kappa^*$ from only observing $F$ and $H$. The following identifiability result has been established:

**Proposition 1.** *(Blanchard et al., 2010) $\kappa^*$ is identifiable if $J$ is irreducible with respect to $H$, that $J$ can not be written as $J = \gamma H + (1 - \gamma)F'$, where $0 \leq \gamma \leq 1$, and $F'$ is another distribution.*

Later, the anchor point condition (Yao et al., 2020b), a stronger requirement was established:

**Proposition 2.** *(Yao et al., 2020b) $\kappa^*$ is identifiable if there exists a subset $S \subseteq \mathcal{Z}$ such that $H(S) > 0$, but $\frac{J(S)}{H(S)} = 0$, where $J(S), H(S)$ denote the probabilities of $S$ measured by $J, H$.*

The above set $S$ is called an anchor set. A sequence of follow-up works have emphasized the necessity of anchor points in identifying a class-dependent transition matrix $T$ (Xia et al., 2019; Li et al., 2021).

Prior work has established the connection between the MPE problem and the learning from noisy label one (Yao et al., 2020b) for the identifiability of an inverse noise rate $\mathbb{P}(Y|\tilde{Y})$ but not the noise transition $T(X)$. We reproduce the discussion and fill in the gap. The discussion and results are for the class-dependent but not instance-dependent label noise, i.e., $T(X) \equiv T$ ($\mathbb{P}(\tilde{Y}|Y, X) \equiv \mathbb{P}(\tilde{Y}|Y)$), and for a binary classification problem. To follow the convention, we assume $Y \in \{-1, +1\}$. There

are two things we need to do: (1) State the noisy label problem as an MPE one; and (2) show that the identifiability of $\kappa^*$ is equivalent to the identifiability of $T$. We start with the first thing above. We want to acknowledge that this equivalence appeared before in (Yao et al., 2020b; Menon et al., 2015). We reproduce it here to make our paper self-contained. Denote by $\pi_+ := \mathbb{P}(Y = -1|\tilde{Y} = +1), \pi_- := \mathbb{P}(Y = +1|\tilde{Y} = -1)$ and $\tilde{\pi}_- = \frac{\pi_-}{1-\pi_+}, \tilde{\pi}_+ = \frac{\pi_+}{1-\pi_-}$.

**Lemma 1.** $\mathbb{P}(X|\tilde{Y} = -1), \mathbb{P}(X|\tilde{Y} = +1)$ *relate to* $\mathbb{P}(X|Y = -1), \mathbb{P}(X|Y = +1)$ *as follows:*

$$\mathbb{P}(X|\tilde{Y} = -1) = \tilde{\pi}_- \cdot \mathbb{P}(X|\tilde{Y} = +1) + (1 - \tilde{\pi}_-) \cdot \mathbb{P}(X|Y = -1) \tag{1}$$

$$\mathbb{P}(X|\tilde{Y} = +1) = \tilde{\pi}_+ \cdot \mathbb{P}(X|\tilde{Y} = -1) + (1 - \tilde{\pi}_+) \cdot \mathbb{P}(X|Y = +1) . \tag{2}$$

Now $\mathbb{P}(X|\tilde{Y} = +1), \mathbb{P}(X|\tilde{Y} = -1)$ correspond to the observed mixture distribution $F, H$, while $\mathbb{P}(X|Y = +1)$ and $\mathbb{P}(X|Y = -1)$ are the two unobserved $J$s, $\tilde{\pi}_-, \tilde{\pi}_+$ correspond to the mixture proportion $\kappa^*$. This has established the learning with noisy label problem as two MPE problems corresponding for the two associated distributions $\mathbb{P}(X|\tilde{Y} = -1), \mathbb{P}(X|\tilde{Y} = +1)$. Therefore to formally establish the equivalence between identifying $\kappa^*$ and $T$, we will only need to establish the equivalence between identifying $\tilde{\pi}_-, \tilde{\pi}_+$ and identifying $T$. Denote by $e_+ := \mathbb{P}(\tilde{Y} = -1|Y = +1), e_- := \mathbb{P}(\tilde{Y} = +1|Y = -1)$ which determine the $T$ for the binary case. We then have:

**Theorem 3.** *Identifying* $\{\tilde{\pi}_-, \tilde{\pi}_+\}$ *is equivalent with identifying* $\{e_-, e_+\}$.

The above theorem concludes the same irreducibility and anchor point conditions proposed under MPE also apply to identifying noise transition matrix $T$. This conclusion aligns with previous successes in estimating class-dependent noise transition matrix $T$ when the anchor point conditions are satisfied (Liu & Tao, 2016; Xia et al., 2019; Li et al., 2021). The above result has **limitations**. Notably, the result focuses on two mixed distributions, leading to the binary classification setup in the noisy learning setting. The authors did not find an easy extension to the multi-class classification problem. Secondly, the translation to the noisy learning problem requires the noise transition matrix to stay the same for a distribution of $X$ (e.g., $\mathbb{P}(X|\tilde{Y} = +1)$), instead of providing instance-level understanding for each $X$.

## 3.2 KRUSKAL'S IDENTIFIABILITY RESULT

Our results build on the Kruskal's identifiability result (Kruskal, 1976; 1977). The setup is as follows: suppose that there is an unobserved variable $Z$ that takes values in a $K$-sized discrete domain $\{1, 2, ..., r\}$. $Z$ has a non-degenerate prior $\mathbb{P}(Z = i) > 0$. Instead of observing $Z$, we observe $p$ variables $\{O_i\}_{i=1}^p$. Each $O_i$ has a finite state space $\{1, 2, ..., \kappa_i\}$ with cardinality $\kappa_i$. Let $M_i$ be a matrix of size $r \times \kappa_i$, which $j$-th row is simply $[\mathbb{P}(O_i = 1|Z = j), ..., \mathbb{P}(O_i = \kappa_i|Z = j)]$. In this case, $[M_1, M_2, ..., M_p]$ and $\mathbb{P}(Z = i)$ are the hidden parameters that control the generation of observations - together, these form our $\theta$. We now introduce the Kruskal rank of a matrix, which plays a central role in Kruskal's identifiability results.

**Definition 3** (Kruskal rank). *(Kruskal, 1976; 1977) For a matrix* $M$, *the Kruskal rank of* $M$ *is the largest number* $I$ *such that every set of* $I$ *rows* [1] *of* $M$ *are linearly independent.*

In this paper, we will use $\mathsf{Kr}(M)$ to denote the Kruskal rank of matrix $M$. To give an example, $M = \begin{bmatrix} 1 & 0 & 0 \\ 0 & 1 & 0 \\ 2 & 0 & 0 \end{bmatrix} \Rightarrow \mathsf{Kr}(M) = 1$. This is because $[1, 0, 0]$ and $[2, 0, 0]$ are linearly dependent. We first reproduce the following theorem:

**Theorem 4.** *(Kruskal, 1976; 1977; Sidiropoulos & Bro, 2000) The parameters* $M_i, i = 1, ..., p$ *are identifiable, up to label permutation, if*

$$\sum_{i=1}^p \mathsf{Kr}(M_i) \geq 2r + p - 1 \tag{3}$$

The result for $p = 3$ was first established in (Kruskal, 1977) demonstrating the power of a three-way tensor, and then it was shown in (Sidiropoulos & Bro, 2000) that the proof extends to a general $p$. The proof builds on showing that different parameter $\theta$ leads to different stacking of $M$s: $[M_1, ..., M_p]$. For example, when $p = 3$, $[M_1, M_2, M_3] := \sum_{k=1}^K \mathbf{m}_1^k \bigotimes \mathbf{m}_2^k \bigotimes \mathbf{m}_3^k$ forms the tensor of the observations, where $\mathbf{m}_i^k, i = 1, 2, 3$ is the $k$-th column of $M_i$.

---

[1] There exists other definition that checks columns. Results would be symmetrical.

## 4 INSTANCE-LEVEL IDENTIFIABILITY

This section will characterize the identifiability of $T(X)$ at the instance level.

### 4.1 SINGLE NOISY LABEL MIGHT NOT BE SUFFICIENT

At a first sight, it is impossible to identify $\mathbb{P}(\tilde{Y}|Y, X)$ from only observing $\mathbb{P}(\tilde{Y}|X)$,[2] unless $X$ satisfies the anchor point definition that $\mathbb{P}(Y = k|X) = 1$ for a certain $k$: since $\mathbb{P}(\tilde{Y}|X) = \mathbb{P}(\tilde{Y}|Y, X) \cdot \mathbb{P}(Y|X)$, different combinations of $\mathbb{P}(\tilde{Y}|Y, X), \mathbb{P}(Y|X)$ can lead to the same $\mathbb{P}(\tilde{Y}|X)$. More specifically, consider the following example:

**Example 1.** *Suppose we have a binary classification problem with* $T(X) = \begin{bmatrix} 1 - e_-(X) & e_-(X) \\ e_+(X) & 1 - e_+(X) \end{bmatrix}$. *Note that using chain rule (probability) we have*

$$\mathbb{P}(\tilde{Y} = +1|X) = \mathbb{P}(\tilde{Y} = +1|Y = +1, X) \cdot \mathbb{P}(Y = +1|X) + \mathbb{P}(\tilde{Y} = +1|Y = -1, X) \cdot \mathbb{P}(Y = -1|X)$$
$$= (1 - e_+(X)) \cdot \mathbb{P}(Y = +1|X) + e_-(X) \cdot \mathbb{P}(Y = -1|X)$$

*Consider two cases: (1):* $\mathbb{P}(Y = +1|X) = 1$, $e_+(X) = e_-(X) = 0.3$ *and (2):* $\mathbb{P}(Y = +1|X) = 0.7$, $e_+(X) = 0.1$, $e_-(X) = 0.233$. *Both cases will return the same* $\mathbb{P}(\tilde{Y} = +1|X) = 0.7$.

Is then the anchor point requirement necessary for identifying $T(X)$ at the instance level? The discussion in the rest of this section departs from the classical single noisy label setting.

### 4.2 THE NECESSITY OF MULTIPLE NOISY LABELS

**Setups** We assume for each instance $X$, we will have $p$ conditionally independent (given $X, Y$) and identically distributed noisy labels $\tilde{Y}_1, ..., \tilde{Y}_p$ generated according to $T(X)$. Let's assume for now we potentially have these labels. Later in this section, we discuss when having multiple redundant labels are possible, and connect to existing solutions in the literature in the next section. For each instance $X$, denote by $K_X \leq K$ the number of non-degenerated label classes $k$ such that $\mathbb{P}(Y = k|X) > 0$. W.l.o.g., let us assume the non-degenerate classes are simply $\{1, 2..., K_X\}$.

Before we formally present the results for having multiple conditionally independent noisy labels, we offer intuitions. The reason behind this identifiability result ties close to latent class model (Clogg, 1995) and tensor decomposition (Anandkumar et al., 2014). When the $p$ noisy labels are conditionally independent given $X$ and $Y$, we will have the joint distribution written as: $\mathbb{P}(\tilde{Y}_1, \tilde{Y}_2, ..., \tilde{Y}_p|Y, X) = \prod_{i=1}^p \mathbb{P}(\tilde{Y}_i|Y, X)$ That is, the joint distribution of noisy labels can be encoded in a much smaller parameter space! In our setup, when we assume the i.i.d. $\tilde{Y}_i, i = 1, 2, ..., p$ are generated according to the same transition matrix $T(X)$, the parameter space is fixed and determined by the size of $T(X)$. Yet, when we increase $p$, the observation space $\mathbb{P}(\tilde{Y}_1, \tilde{Y}_2, ..., \tilde{Y}_p|Y, X)$ becomes richer to help us identify $T(X)$. We now define an *informative noisy label*.

**Definition 4.** *For a given* $(X, Y)$, *we call their noisy label* $\tilde{Y}$ *informative if* $rank(T(X)) = K_X$.

Definition 4 requires the $K_X$ rows of $T(X)$ are linearly independent. When the observation space for $\tilde{Y}$ is the same as $Y$ (therefore $T(X)$ is a squared matrix), i.e., the true label $Y$ has a full support on the entire label space, then the requirement is stating that $T(X)$ is of full rank, which is already assumed in the literature - e.g., loss correction (Natarajan et al., 2013; Patrini et al., 2017a) would require the matrix has an inverse $T^{-1}(X)$, which is equivalent to $T(X)$ being full rank. In particular, it was required $e_+(X) + e_-(X) < 1$ in (Natarajan et al., 2013), which can be easily shown to imply $T(X)$ is full rank. But we don't remove the possibility that $T(X)$ is not a squared matrix and $K_X$ can be much smaller than the entire label space. Our first identifiability result states as follows:

**Theorem 5.** *With i.i.d. noisy labels, three informative noisy labels* $\tilde{Y}_1, \tilde{Y}_2, \tilde{Y}_3$ *($p = 3$) are both sufficient and necessary to identify* $T(X)$ *when* $K_X \geq 2$.

Note that $K_X \geq 2$ is easily satisfied as long as there exists uncertainty in $\mathbb{P}(Y|X)$.

---

[2]We clarify that we will require knowing $\mathbb{P}(\tilde{Y}|X)$ - this requirement may appear weird when only one noisy label is sampled. But in practice, there are tools available to regress the posterior function $\mathbb{P}(\tilde{Y}|X)$ for each $X$.

*Proof sketch.* We provide the key steps of the proof. The full proof can be found in the supplemental material. We first prove sufficiency. We first relate our problem setting to the setup of Kruskal's identifiability scenario: $Y \in \{1, 2, ..., K_X\}$. corresponds to the unobserved hidden variable $Z$. $\mathbb{P}(Y = i)$ corresponds to the prior of this hidden variable. Each $\tilde{Y}_i, i = 1, ..., p$ corresponds to the observation $O_i$. $\kappa_i$ is then simply the cardinality of the noisy label space, $K$. In the context of this theorem, $p = 3$, corresponds to the three noisy labels we have. Each $\tilde{Y}_i$ corresponds to an observation matrix $M_i$: $M_i[j, k] = \mathbb{P}(O_i = k | Z = j) = \mathbb{P}(\tilde{Y}_i = k | Y = j, X)$. Therefore, by definition of $M_1, M_2, M_3$ and $T(X)$, they all equal to $T(X)$: $M_i \equiv T(X), i = 1, 2, 3$. When $T(X)$ has a rank $K_X$, we know immediately that all rows in $M_1, M_2, M_3$ are independent. Therefore, the Kruskal ranks satisfy $\mathsf{Kr}(M_1) = \mathsf{Kr}(M_2) = \mathsf{Kr}(M_3) = K_X$. Checking the condition in Theorem 4, we easily verify $\mathsf{Kr}(M_1) + \mathsf{Kr}(M_2) + \mathsf{Kr}(M_3) = 3K_X \geq 2K_X + 2$ . Calling Theorem 4 proves the sufficiency.

To prove necessity, we need to prove less than 3 informative labels will not suffice to guarantee identifiability. The idea is to show that the two different sets of parameters $T(X)$ can lead to the same joint distribution $\mathbb{P}(\tilde{Y}_1, \tilde{Y}_2 | X)$. We leave the detailed constructions to the supplemental material. □

The above result points out that to ensure identifiability of $T(X)$ at the instance level, we would need three conditionally independent and informative noisy labels. This result coincides with a couple of recent works that promote the use of three redundant labels (Liu et al., 2020; Zhu et al., 2021c; Zhang et al., 2014). Per our theorem, these two proposed solutions have a more profound connection to the identifiability of hidden parametric models, and three labels are not only algorithmically sufficiently, but also necessary. This result also echoes the power of tensor (stacking third order information) in uncovering hidden models (Traganitis et al., 2018; Zhang et al., 2014). Particularly relevant to us is (Zhang et al., 2014) where it was shown a spectral EM approach that uses three noisy labels suffices to identify the noise transition matrix of labels. We want to highlight that our proof and results establish both the necessity and sufficiency for having three informative noisy labels, independent from the specific algorithms developed. Another note we want to add is that our main inquiry is on establishing the conditions for identifying $T(X)$, instead of proposing algorithms to estimate $T(X)$.

The crowdsourcing community has been largely focusing on soliciting more than one label from crowdsourced workers, yet the learning from noisy label literature has primarily focused on learning from a single one. One of the primary motivations of crowdsourcing multiple noisy labels is indeed to aggregate them into a cleaner one (Liu et al., 2012; Karger et al., 2011; Liu & Liu, 2015), which serves as a pre-processing step towards solving the noisy learning problem. Nonetheless, our result demonstrates the other significance of having multiple labels - they help the learner identify the underlying true noise transition parameters.

## 5    INSTANTIATIONS AND PRACTICAL IMPLICATIONS OF OUR RESULTS

Most of the learning with noisy label solutions focus on the case of using a single label and have observed empirical successes. In this section, we provide extensions of our results to cover of state-of-the-art learning with noisy label methods, together with specific assumptions over $X, T(X) = [\mathbb{P}(\tilde{Y}|Y, X)]$ etc. We show that our results can easily extend to these specific instantiations that successfully avoided the requirements of having multiple noisy labels for each $X$. The high-level intuition for Section 5.1 is to leverage the smoothness and clusterability of the nearest neighbor $X$s so that their noisy labels will jointly serve as the multiple noisy labels for the local group. Section 5.2 and 5.3 build on the notion that if $T(X)$ is the same for a group of $X$s, each group can then be treated as one "instance" and a "disentangled" version of $X$ will become observation variables that serve the similar role of the additionally required noisy labels.

### 5.1    LEVERAGING SMOOTHNESS AND CLUSTERABILITY OF $X$

We start with a discussion using the smoothness and clusterability of $X$. Recent results have explored the clusterability of $X$s (Zhu et al., 2021c; Bahri et al., 2020) to infer the noise transition matrix:

**Definition 5.** *The 2-NN clusterability requires each $X$ and its two nearest neighbors $X_1, X_2$ share the same true label $Y$, that is $Y = Y_1 = Y_2$, and $T(X) = T(X_1) = T(X_2)$.*

This definition helps us remove the requirment for multiple noisy labels per each $X$: one can view it as for each $X$, borrowing the noisy labels from its 2-NN, we have three independent noisy labels $\tilde{Y}, \tilde{Y}_1, \tilde{Y}_2$, all from the same $Y$. This smoothness or clusterability condition allows us to apply our

identifiability results when one believes the $T(X)$ stays the same for the 2-NN nearest neighborhood $X, X_1, X_2$. But, when does an instance $X$ and its 2-NN $X_1, X_2$ share the same true label? This requirement seems strange at the first sight: as long as $\mathbb{P}(Y|X), \mathbb{P}(Y_1|X_1)$ are not degenerate (being either 0 or 1 for different label classes), there always seems to be a positive probability that the realized $Y \neq Y_1$, no matter how close $X$ and $X_1$ are. Nonetheless, the 2-NN requirement seems to hold empirically: according to (Zhu et al., 2021c) (Table 3 therein), when using a feature extractor built using the clean label, more than 99% of the instance satisfies the 2-NN condition. Even when using a weaker feature extractor, the ratio is mostly always in or close to the $80\%$ range.

The following data generation process for an unstructured discrete domain of classification problems (Feldman, 2020; Liu, 2021) helps us justify the 2-NN requirement. The intuition is that when $X$s are informative and sufficiently discriminative, the similar $X$s are going to enjoy the same true label.

- Let $\lambda = \{\lambda_1, ..., \lambda_n\}$ denote the priors for each $X \in \mathcal{X}$.
- For each $X \in \mathcal{X}$, sample a quantity $q_X$ independently and uniformly from the set $\lambda$.
- The resulting probability mass function of $X$ is given by $D(X) = \frac{q_X}{\sum_{X \in \mathcal{X}} q_X}$.
- A total of $N$ $X$s are observed. Denote by $X_1, X_2$ $X$'s two nearest neighbors.
- Each $(X, X_1, X_2)$ forms a triplet if $||X_1 - X||, ||X_2 - X||$ fall below a threshold $\epsilon$ (closeness).
- A single $Y$ for the tuple $(X, X_1, X_2)$ draws from $\mathbb{P}(Y|X, X_1, X_2)$.
- Based on $Y$, we further observe three $\tilde{Y}, \tilde{Y}_1, \tilde{Y}_2$ according to $\mathbb{P}(\tilde{Y}, \tilde{Y}_1, \tilde{Y}_2|Y)$.

The above data-generation process captures the correlation among $X$s that are really close. We prove the above data generation process satisfies the 2-NN clusterability requirement with high probability.

**Theorem 6.** *When $N$ is large enough such that $N > \frac{4 \sum_{X \in \mathcal{X}} q_X}{\min_X q_X}$, w.p. at least $1 - N exp(-2N)$, each $X$ and its two nearest neighbor $X_1, X_2$ satisfy the 2-NN clusterability.*

**Smoothness conditions in semi-supervised learning** This above discussion also ties closely to the smoothness requirements in semi-supervised learning (Zhu et al., 2003; Zhu, 2005), where the neighborhood $X$s can provide and propagate label information in each local neighborhood of $X$s. Indeed, this idea echoes the co-teaching solution (Jiang et al., 2018; Han et al., 2018) in the literature of learning with noisy labels, where a teacher/mentor network is trained to provide artificially generated noisy labels to supervise the training of the student network. Our identifiability result, to a certain degree, implies that the addition of the additional noisy supervision improves the chance for identifying $T(X)$. In (Jiang et al., 2018; Han et al., 2018), counting the noisy label itself, and the "teacher" supervision, there are two such noisy supervision labels. This observation raises an interesting question: since our result emphasized three labels, does adding an additional teacher network for an additional supervision help? This question merits empirical verification.

## 5.2 Leveraging smoothness and clusterability of $T(X)$

We show that another "smoothness" assumption of $T(X)$ introduces new observation variables for us to identify $T(X)$. In Figure 1, we define variable $G = \{1, 2, ..., |G|\}$ to denote the group membership for each $X$. Consider a scenario that $X$ can be grouped into $|G|$ groups such that each group of $X$s share the same $T(X)$: $T(X_1) = T(X_2)$ if $X_1, X_2$ share the same group membership. We observe $G, X, \tilde{Y}$. This type of grouping has been observed in the literature:

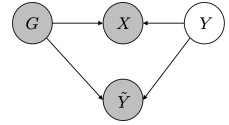

Figure 1: Graph for $(G, X, Y, \tilde{Y})$. Grey color indicates observable variables.

**Class-dependent** $T$ $\mathbb{P}(\tilde{Y}|Y, X) \equiv \mathbb{P}(\tilde{Y}|Y)$, a single group $\forall X$s.

**Noise clusterability** The noise transition estimator proposed in (Zhu et al., 2021c) was primarily developed for class-dependent but not instance-dependent $T(X)$. Nonetheless, a noise clusterability definition is introduced therein to allow the approach to be applied to instance-dependent noise. Under noise clusterability, using clustering algorithms can help separate the dataset into local ones.

**Group-dependent** $T(X)$ Recent results have also studied the case that the data $X$ can be grouped using additional information (Wang et al., 2021a; Liu & Wang; Wang et al., 2021b). For instance, (Wang et al., 2021a; Liu & Wang) consider the setting where the data can be grouped by the associated

"sensitive information", e.g., by age, gender, or race. Then the noise transition matrix remains the same for $X$s that come from each group.

By this grouping, $X$ becomes informative observations for each hidden $Y$ and will fulfill the requirement of observing additional noisy labels. We now define a disentangled feature and an informative feature: Denote by $R(X) \in \mathbb{R}^{d^*}$ a learned representation for $X$. Denote by $R_i$ the random variable for $R_i(X), i = 1, 2, .., d^*$. For simplicity of the analysis, we assume each $R_i$ has finite observation space $\mathcal{R}_i$ with cardinality $|\mathcal{R}_i| = \kappa_i$. Define $M_i$ for each $R_i$ as $M_i[j, k] = \mathbb{P}(R_i = \mathcal{R}_i[k]|Y = j)$, where in above $\mathcal{R}_i[k]$ denotes the $k$-th element in $\mathcal{R}_i$.

**Definition 6** (Disentangled $R$). *$R$ is disentangled if $\{R_i\}_{i=1}^{d^*}$ are conditional independent given $Y$.*

**Definition 7** (Informative features). *$R_i$ is informative if its Kruskal rank is at least 2: $\mathsf{Kr}(M_i) \geq 2$.* Assuming each $X$ can be transformed into a set of disentangled features $R$, we prove:

**Theorem 7.** *For $X$s in a given group $g \in G$, with a single informative noisy label, $T(X)$ is identifiable if the number of disentangled and informative features $d^*$ satisfy that $d^* \geq K$.*

This result points out a new observation that even when we have a single noisy label, given a sufficient number of disentangled and informative features, the noise transition matrix $T$ is indeed identifiable, without requiring either multiple noisy labels, or the anchor point condition. The above result aligns with recent discussions of a neural network being able to disentangle features (Higgins et al., 2018; Steenbrugge et al., 2018) proves to be a helpful property. We establish that having disentangled feature helps identify $T(X)$. The required number of disentangled features grows linearly in $K$. When relaxing the unique identifiability to generic identifiability, i.e., the identifiability scenario has measure zero (Allman et al., 2009), the above theorem can be further extended to requiring $d^* \geq \lceil \log_2 \frac{2K_G^*+1}{2} \rceil$, where $K_G^* = \max_{X \in G} K_X$. Details are deferred to Appendix (Theorem 10). Note that the existence of disentangled $X$ does not imply that we will be able to directly infer $\mathbb{P}(Y|X)$ which will help us complete the learning task directly. But rather, it is indeed possible to further identify the structure $\mathbb{P}(X|Y)$ (from unobserved to observed) but this is an identifiability problem defined on a much higher space.

When disentangled features are not given, how do we disentangle $X$ using only noisy labels to benefit from our results? In Section 6 we will test the effectiveness of a self-supervised representation learning approach that takes the side information relative to true label $Y$ but operates independently from noisy labels. This result implies when the noise rate is high such that $\tilde{Y}$ starts to become uninformative, dropping the noisy labels and focusing on obtaining the disentangled features helps with the identifiability of $T(X)$. This observation also helps explain successes in applying semi-supervised (Cheng et al., 2021a; Li et al., 2020; Nguyen et al., 2019) and self-supervised learning (Cheng et al., 2021b; Zheltonozhskii et al., 2022; Ghosh & Lan, 2021) to handle noisy labels.

## 5.3 SMOOTHNESS AND CLUSTERABILITY OF $T(X)$ WITH UNKNOWN GROUPINGS

In practice, we often do not know the groupings of $X$ that share the same $T(X)$, nor do we have a clear power (e.g., the noise clusterability condition) to separate the data into different groups. In reality, different from Figure 1, the group membership can often remain hidden, if no additional knowledge of the data is solicited, leading to a situation in Figure 2. It is a non-trivial task to jointly infer the group membership with $T(X)$. We first show that mixing the group membership can lead to non-negligible estimation errors. Suppose that there are two groups of $X$, each having a noise transition matrix $T_1(X), T_2(X)$. Suppose we ended up estimating one $T^*(X)$ for both groups mistakenly. We then have:

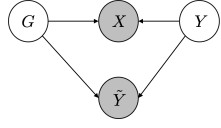

Figure 2: Graph with unobserved $G$. Grey color indicates observable variables.

**Theorem 8.** *Any estimator $T^*(X)$ will incur at least the following estimation error:*

$$||T_1(X) - T^*(X)||_F + ||T_2(X) - T^*(X)||_F \geq (1/\sqrt{2}) \cdot ||T_1(X) - T_2(X)||_F \quad (4)$$

The above result shows the necessity of identifying $G$ as well. Now we present our positive result on the identifiability when $G$ is hidden too: Re-number the combined space of $G \times Y$ as $\{1, 2, ..., |G|K\}^3$. We are going to reuse the definition of $M_i$ for each disentangled feature $R_i$: Define the "Kruskal matrix" for each $R_i$ as $M_i[j, k] = \mathbb{P}(R_i = \mathcal{R}_i[k]|G \times Y = j)$.

---

[3]By mapping $(G = 1, Y = 1) \to 1, (G = 1, Y = 2) \to 2, ..., (G = |G|, Y = K) \to |G|K$.

**Theorem 9.** *For $Xs$ in a given group $g \in G$, with a single informative noisy label, $T(X)$ is identifiable if the number of disentangled and informative features $d^*$ satisfy that $d^* \geq 2|G|K - 1$.* When we have unknown groups of noise, the requirement of the number of informative and disentangled features grows linearly in $|G|$. We now relate to the literature that implicitly groups $Xs$. We will use $\mathcal{X}$ to denote the space of all possible $Xs$.

**Part-dependent label noise** (Xia et al., 2020b) discusses a part-dependent label noise model where each $T(X)$ can decompose into a linear combination of $p$ parts: $T(X) = \sum_i^p \omega_i(X) \cdot T_i$. The motivation of the above model is each $X$ can be viewed as a combination of multiple different sub-parts, and each of them has a certain difficulty being labeled. The hope is that the parameter space $\omega(X)$ can reduce the dependency between $X$ and $T(X)$. Denote $\mathcal{W} := \{\omega(X) : X \in \mathcal{X}\}$. To put into our result, $|G| = |\mathcal{W}|$. If $\mathcal{W}$ has a much smaller space than $\mathcal{X}$, the condition specified in Theorem 9 would be more likely to be satisfied.

**DNN approach** (Yang et al., 2021) proposes using a deep neural network to encode the dependency between $X$ and $T^*(X)$, with the only difference being that $T^*(X)$ is defined as the transition between $\tilde{Y}$ and the Bayes optimal label $Y^*$. Define: $\text{DNN} := \{\text{DNN}(X) : X \in \mathcal{X}\}$. Similarly, in analogy to our results in Theorem 9, with replacing the hidden variable $Y$ to $Y^*$, $|G|$ will be determined by $|\text{DNN}|$. So long as the DNN can identify the patterns in $T(X)$ and compress the space of $\text{DNN}(X)$ as compared to $\mathcal{X}$, the identifiability becomes easier to achieve.

**The causal approach** (Yao et al., 2021) proposed improving the identifiability by exploring the causal structure. With causal inference, one can identify a more representative and compressed $\tilde{X}$ for each $X$ such that $\mathbb{P}(\tilde{Y}|Y, X, \tilde{X}) = \mathbb{P}(\tilde{Y}|Y, \tilde{X})$. Denote $\tilde{\mathcal{X}} := \{\tilde{X} : \tilde{X} \to X \in \mathcal{X}\}$, and $|G| = |\tilde{\mathcal{X}}|$.

## 6 Some Empirical Evidence: Disentangled Features

Most of our results above verified the empirical success of existing approaches from the identifiability's perspective and we refer the interested reader to the detailed experiments in the corresponding references. We now empirically show the possibility of learning disentangled features to help identify the noise transition matrix. We consider three types of encoders that are used to generate features. The first encoder is pre-trained by cross-entropy (CE) loss via a weakly supervised manner which is generally adopted in FW (Patrini et al., 2017b) and HOC (Zhu et al., 2021c). However, since the training data is noisy, it is hard to guarantee that features are disentangled - this is our baseline. The second encoder is pre-trained by SimCLR (Chen et al., 2020) via a self-supervised manner. It is shown that the features trained by SimCLR are partly disentangled on some simple augmentation features such as rotation and colorization (Wang et al., 2021c). The third encoder is trained by IPIRM (Wang et al., 2021c) via a self-supervised manner which can generate fully disentangled features. After training these three encoders, we fix the encoder and generate features from raw samples to estimate the noise transition matrix using HOC estimator (Zhu et al., 2021c). We evaluate the performance via absolute estimation error defined below: $\text{err} = \sum_{i=1}^K \sum_{j=1}^K |\hat{T}_{i,j} - T_{i,j}|/K^2 \cdot 100$, where $\hat{T}$ is the estimated noise transition matrix, $T$ is the real noise-transition matrix, $K$ is the number of classes in the dataset, which is also the size of the transition matrix. The overall experiments are shown in Table 1. We observe that the estimation error decreases as features become more disentangled which supports our analyses. We defer the details, more experiments, as well as experiments on comparing training performances using disentangled features, to the supplementary material.

Table 1: Comparison of estimation error for different types of features on CIFAR-10. Each experiment is run 3 times and mean $\pm$ std is reported. *asymm.*: asymmetric label noise; *inst.*: instance-dependent label noise. Numbers are noise rates. All the encoders are from ResNet50 backbone.

| Feature Type | asymm. 0.3 | asymm. 0.4 | inst. 0.4 | inst. 0.5 | inst. 0.6 |
|---|---|---|---|---|---|
| Weakly-Supervised | $14.51 \pm 0.4$ | $15.2 \pm 0.02$ | $8.39 \pm 0.05$ | $6.91 \pm 0.06$ | $6.18 \pm 0.15$ |
| SimCLR | $4.42 \pm 0.01$ | $4.41 \pm 0.01$ | $2.91 \pm 0.02$ | $2.55 \pm 0.04$ | $2.64 \pm 0.03$ |
| IPIRM | $\mathbf{3.73 \pm 0.02}$ | $\mathbf{3.74 \pm 0.01}$ | $\mathbf{2.47 \pm 0.03}$ | $\mathbf{2.20 \pm 0.02}$ | $\mathbf{2.37 \pm 0.06}$ |

**Concluding remarks** This paper characterizes the identifiability of instance-level label noise transition matrix. We connect the problem to the celebrated Kruskal's identifiability result and present a necessary and sufficient condition for the instance-level identifiability. We extend and instantiate our results to practical settings to explain the successes of existing solutions. We show the importance of disentangled and informative features for identifying the noise transition matrix.

## 7 ETHICAL STATEMENT

We are not aware of negative societal consequence of applying our results. But at multiple places of the work, we state the limitations of the setup to not mislead readers into misunderstanding our claims. For instance, we discussed the situation when we will have multiple noisy labels and our focus on discretized features. In Section 5.2, we clearly stated our requirement of the disentangled and informative features.

## 8 REPRODUCIBILITY STATEMENT

We include the following checklist for the purpose of reproducibility:

1. Do the main claims made in the abstract and introduction accurately reflect the paper's contributions and scope? [Yes]

2. If you are including theoretical results...

   (a) Did you state the full set of assumptions of all theoretical results? [Yes]
   (b) Did you include complete proofs of all theoretical results? [Yes] We present the complete proofs in the appendix and have added detailed explanations.

3. If you ran experiments...

   (a) Did you include the code, data, and instructions needed to reproduce the main experimental results (either in the supplemental material or as a URL)? [Yes] We submitted experiment details in the appendix and the implementations in the supplementary materials.
   (b) Did you specify all the training details (e.g., data splits, hyperparameters, how they were chosen)? [Yes]
   (c) Did you report error bars (e.g., with respect to the random seed after running experiments multiple times)? [Yes]
   (d) Did you include the total amount of compute and the type of resources used (e.g., type of GPUs, internal cluster, or cloud provider)? [Yes]

4. If you are using existing assets (e.g., code, data, models) or curating/releasing new assets...

   (a) If your work uses existing assets, did you cite the creators? [Yes]
   (b) Did you mention the license of the assets? [Yes]
   (c) Did you include any new assets either in the supplemental material or as a URL? [No]
   (d) Did you discuss whether and how consent was obtained from people whose data you're using/curating? [N/A]
   (e) Did you discuss whether the data you are using/curating contains personally identifiable information or offensive content? [N/A]

5. If you used crowdsourcing or conducted research with human subjects...

   (a) Did you include the full text of instructions given to participants and screenshots, if applicable? [N/A]
   (b) Did you describe any potential participant risks, with links to Institutional Review Board (IRB) approvals, if applicable? [N/A]
   (c) Did you include the estimated hourly wage paid to participants and the total amount spent on participant compensation? [N/A]

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

# APPENDIX: IDENTIFIABILITY OF LABEL NOISE TRANSITION MATRIX

The Appendix is organized in the following way: Section A proves the Theorems in the main paper; Section B provides more discussions on generic identifiability; Section C provides more experiments on learning with noisy labels *w.r.t.* disentangled features and elaborates the detailed experimental settings in the paper.

## A OMITTED PROOFS

### PROOF FOR LEMMA 1

*Proof.* Using Bayes rule we easily obtain

$$
\begin{aligned}
\mathbb{P}(X|\tilde{Y}=+1) = \mathbb{P}(X|Y=+1)\cdot\mathbb{P}(Y=+1|\tilde{Y}=+1) \\
+ \mathbb{P}(X|Y=-1)\cdot\mathbb{P}(Y=-1|\tilde{Y}=+1)
\end{aligned}
\tag{5}
$$

The equality is due to the fact that $\tilde{Y}$ and $X$ are assumed to be independent given $Y$. Similarly:

$$
\begin{aligned}
\mathbb{P}(X|\tilde{Y}=-1) = \mathbb{P}(X|Y=+1)\cdot\mathbb{P}(Y=+1|\tilde{Y}=-1) \\
+ \mathbb{P}(X|Y=-1)\cdot\mathbb{P}(Y=-1|\tilde{Y}=-1)
\end{aligned}
\tag{6}
$$

Since both $\mathbb{P}(X|Y=+1), \mathbb{P}(X|Y=-1)$ are unknown, solving Eqn. (5) and (6) we further have

$$
\mathbb{P}(X|\tilde{Y}=-1) = \tilde{\pi}_- \cdot \mathbb{P}(X|\tilde{Y}=+1) + (1-\tilde{\pi}_-) \cdot \mathbb{P}(X|Y=-1)
\tag{7}
$$

$$
\mathbb{P}(X|\tilde{Y}=+1) = \tilde{\pi}_+ \cdot \mathbb{P}(X|\tilde{Y}=-1) + (1-\tilde{\pi}_+) \cdot \mathbb{P}(X|Y=+1).
\tag{8}
$$

$\square$

### PROOF FOR THEOREM 3

*Proof.* Further from $\tilde{\pi}_-, \tilde{\pi}_+$ we can solve and derive $\pi_- = \frac{\tilde{\pi}_-(1-\tilde{\pi}_+)}{1-\tilde{\pi}_-\tilde{\pi}_+}, \pi_+ = \frac{\tilde{\pi}_+(1-\tilde{\pi}_-)}{1-\tilde{\pi}_-\tilde{\pi}_+}$, establishing the equivalence between identifying $\tilde{\pi}_-, \tilde{\pi}_+$ with identifying $\pi_-, \pi_+$. Next we show that identifying $\pi_-, \pi_+$ is equivalent with identifying $\{e_+, e_-\}$.

We first show identifying $\{\pi_+, \pi_-\}$ suffices to identify $\{e_+, e_-\}$. To see this,

$$
\mathbb{P}(\tilde{Y}=+1|Y=-1) = \frac{\mathbb{P}(Y=-1|\tilde{Y}=+1)\mathbb{P}(\tilde{Y}=+1)}{\mathbb{P}(Y=-1)}
$$

And:

$$
\mathbb{P}(Y=-1) = \mathbb{P}(Y=-1|\tilde{Y}=+1)\mathbb{P}(\tilde{Y}=+1) + \mathbb{P}(Y=-1|\tilde{Y}=-1)\mathbb{P}(\tilde{Y}=-1)
$$

The derivation for $\mathbb{P}(\tilde{Y}=-1|Y=+1)$ is entirely symmetric. Since we directly observe $\mathbb{P}(\tilde{Y}=-1), \mathbb{P}(\tilde{Y}=+1)$, with identifying $\mathbb{P}(Y=+1|\tilde{Y}=-1), \mathbb{P}(Y=-1|\tilde{Y}=+1)$, we can identify $\mathbb{P}(\tilde{Y}=+1|Y=-1), \mathbb{P}(\tilde{Y}=-1|Y=+1)$.

Next we show that to identify $\{e_+, e_-\}$, it is necessary to identify $\{\pi_+, \pi_-\}$. Suppose not: we are unable to identify $\pi_+, \pi_i$ but are able to identify $\{e_+, e_-\}$. This implies that there exists another pair $\{\pi'_+, \pi'_-\} \neq \{\pi_+, \pi_-\}$ such that (denote by $\tilde{p} := \mathbb{P}(\tilde{Y}=+1)$)

$$
\mathbb{P}(\tilde{Y}=+1|Y=-1) = \frac{\pi_+\tilde{p}}{\pi_+\tilde{p} + (1-\pi_-)(1-\tilde{p})}
\tag{9}
$$

$$
= \frac{\pi'_+\tilde{p}}{\pi'_+\tilde{p} + (1-\pi'_-)(1-\tilde{p})}
\tag{10}
$$

$$\mathbb{P}(\tilde{Y} = -1|Y = +1) = \frac{\pi_-(1-\tilde{p})}{(1-\pi_+)\tilde{p} + \pi_-(1-\tilde{p})} \tag{11}$$

$$= \frac{\pi'_-(1-\tilde{p})}{(1-\pi'_+)\tilde{p} + \pi'_-(1-\tilde{p})} \tag{12}$$

By dividing $\pi_+, \pi'_+$ in both the numerator and denominator in Eqn. (9) and (10), we conclude that

$$\frac{1-\pi_-}{\pi_+} = \frac{1-\pi'_-}{\pi'_+} \tag{13}$$

While from Eqn. (11) and (12) we conclude

$$\frac{1-\pi_+}{\pi_-} = \frac{1-\pi'_+}{\pi'_-} \tag{14}$$

From Eqn. (13) and (14) we have

$$(1-\pi_-)\pi'_+ = (1-\pi'_-)\pi_+ \tag{15}$$

$$(1-\pi'_+)\pi_- = (1-\pi_+)\pi'_- \tag{16}$$

Taking the difference and re-arrange terms we prove

$$\pi_+ + \pi_- = \pi'_+ + \pi'_-$$

From Eqn. (13) again, taking $-1$ on both side we have

$$\frac{1-\pi_- - \pi_+}{\pi_+} = \frac{1-\pi'_- - \pi'_+}{\pi'_+} \tag{17}$$

This proves $\pi_+ = \pi'_+$. Similarly we have $\pi_- = \pi'_-$ - but this contradicts the assumption that $\{\pi'_-, \pi'_+\}$ is a different pair. $\qquad\square$

## PROOF FOR THEOREM 5

*Proof.* We first prove sufficiency. We first relate our problem setting to the setup of Kruskal's identifiability scenario: $Y \in \{1, 2, ..., K_X\}$ corresponds to the unobserved hidden variable $Z$. $\mathbb{P}(Y = i)$ corresponds to the prior of this hidden variable. Each $\tilde{Y}_i, i = 1, ..., p$ corresponds to the observation $O_i$. $\kappa_i$ is then simply the cardinality of the noisy label space, $K$. In the context of this theorem, $p = 3$, corresponding to the three noisy labels we have.

Each $\tilde{Y}_i$ corresponds to an observation matrix $M_i$:

$$M_i[j, k] = \mathbb{P}(O_i = k|Z = j) = \mathbb{P}(\tilde{Y}_i = k|Y = j, X)$$

Therefore, by definition of $M_1, M_2, M_3$ and $T(X)$, they all equal to $T(X)$: $M_i \equiv T(X), i = 1, 2, 3$. When $T(X)$ has a rank $K_X$, we know immediately that all rows in $M_1, M_2, M_3$ are linearly independent. Therefore, the Kruskal ranks satisfy

$$\mathsf{Kr}(M_1) = \mathsf{Kr}(M_2) = \mathsf{Kr}(M_3) = K_X$$

Checking the condition in Theorem 4, we easily verify

$$\mathsf{Kr}(M_1) + \mathsf{Kr}(M_2) + \mathsf{Kr}(M_3) = 3K_X \geq 2K_X + 2$$

Calling Theorem 4 proves the sufficiency.

Now we prove necessity. To prove so, we are allowed to focus on the binary case, where

$$T(X) = \begin{bmatrix} 1 - e_-(X) & e_-(X) \\ e_+(X) & 1 - e_+(X) \end{bmatrix}$$

Note in above, for simplicity we drop $e_-, e_+$'s dependency in $X$. We need to prove less than 3 informative labels will not suffice to guarantee identifiability. The idea is to show that the two different set of parameters $e_-, e_+$ can lead to the same joint distribution $\mathbb{P}(\tilde{Y}_1, \tilde{Y}_2|X)$.

The case with a single label is already proved by Example 1. Now consider two noisy labels $\tilde{Y}_1, \tilde{Y}_2$. We first claim the following three quantities fully capture the information provided by $\tilde{Y}_1, \tilde{Y}_2$:

- Posterior: $\mathbb{P}(\tilde{Y}_1 = +1|X)$

- Positive Consensus: $\mathbb{P}(\tilde{Y}_1 = \tilde{Y}_2 = +1|X)$

- Negative Consensus: $\mathbb{P}(\tilde{Y}_1 = \tilde{Y}_2 = -1|X)$

This is because other statistics in $\tilde{Y}_1, \tilde{Y}_2|X$ can be reproduced using combinations of the three quantities above:

$$\mathbb{P}(\tilde{Y}_1 = -1|X) = 1 - \mathbb{P}(\tilde{Y}_1 = +1|X) \,,$$
$$\mathbb{P}(\tilde{Y}_1 = +1, \tilde{Y}_2 = -1|X) = \mathbb{P}(\tilde{Y}_1 = +1|X) - \mathbb{P}(\tilde{Y}_1 = \tilde{Y}_2 = +1|X) \,,$$
$$\mathbb{P}(\tilde{Y}_1 = -1, \tilde{Y}_2 = +1|X) = \mathbb{P}(\tilde{Y}_2 = +1|X) - \mathbb{P}(\tilde{Y}_1 = \tilde{Y}_2 = +1|X) \,.$$

But $\mathbb{P}(\tilde{Y}_2 = +1|X) = \mathbb{P}(\tilde{Y}_1 = +1|X)$, since the two noisy labels are identically distributed. The above three quantities led to three equations that depend on $e_+, e_-$: denote by $\gamma := \mathbb{P}(Y = +1)$

Next we prove the following system of equations:

$$\mathbb{P}(\tilde{Y} = +1|X) = \gamma \cdot (1 - e_+) + (1 - \gamma) \cdot e_-$$
$$\mathbb{P}(\tilde{Y}_1 = \tilde{Y}_2 = +1|X) = \gamma \cdot (1 - e_+)^2 + (1 - \gamma) \cdot e_-^2$$
$$\mathbb{P}(\tilde{Y}_1 = \tilde{Y}_2 = -1|X) = \gamma \cdot e_+^2 + (1 - \gamma) \cdot (1 - e_-)^2$$

To see this:

$$\mathbb{P}(\tilde{Y}_1 = \tilde{Y}_2 = +1|X)$$
$$= \mathbb{P}(\tilde{Y}_1 = \tilde{Y}_2 = +1, Y = +1|X)$$
$$\quad + \mathbb{P}(\tilde{Y}_1 = \tilde{Y}_2 = +1, Y = -1|X)$$
$$= \mathbb{P}(\tilde{Y}_1 = \tilde{Y}_2 = +1|Y = +1, X) \cdot \mathbb{P}(Y = +1|X)$$
$$\quad + \mathbb{P}(\tilde{Y}_1 = \tilde{Y}_2 = +1|Y = -1, X) \cdot \mathbb{P}(Y = -1|X)$$
$$= \gamma \cdot (1 - e_+)^2 + (1 - \gamma) \cdot e_-^2$$

The last equality uses the fact that $\tilde{Y}_1, \tilde{Y}_2$ are conditional independent given $Y$, so

$$\mathbb{P}(\tilde{Y}_1 = \tilde{Y}_2 = +1|Y = +1, X) =$$
$$\mathbb{P}(\tilde{Y}_1 = +1|Y = +1, X) \cdot \mathbb{P}(\tilde{Y}_2 = +1|Y = +1, X)$$
$$\mathbb{P}(\tilde{Y}_1 = \tilde{Y}_2 = +1|Y = -1, X) =$$
$$\mathbb{P}(\tilde{Y}_1 = +1|Y = -1, X) \cdot \mathbb{P}(\tilde{Y}_2 = +1|Y = -1, X)$$

We can similarly derive for $\mathbb{P}(\tilde{Y}_1 = \tilde{Y}_2 = -1|X)$.

Now we show the above equations do not identify $e_+, e_-$. For instance, it is straightforward to verify that both of the solutions below satisfy the equations (up to numerical errors, exact solution exists but in complicated forms):

- $\gamma = 0.7$, $e_+ = 0.2$, $e_- = 0.2$

- $\gamma = 0.8$, $e_+ = 0.242$, $e_- = 0.07$

The above example proves that two informative noisy labels are insufficient to guarantee identifiability.

For completeness we provide rationals for the multi-class case too. The idea is to show that the complete information returned by the single noisy label and two noisy labels do not always guarantee a unique solution.

For the first order information:

$$\mathbb{P}(\tilde{Y} = i|X) = \sum_{k \in [K]} \mathbb{P}(Y = k) \cdot \mathbb{P}(\tilde{Y} = i|Y = k, X)$$

$$= \sum_{k \in [K]} \mathbb{P}(Y = k|X) \cdot T_{ki}(X)$$

Enumerating all $i$s, there are $K$ equations, written in a matrix form as:

$$\tilde{\mathbf{P}} = (T(X))^\top \cdot \mathbf{P}$$

where $\tilde{\mathbf{P}}$ is the vector form for $[\mathbb{P}(\tilde{Y} = 1|X); \mathbb{P}(\tilde{Y} = 2|X); ...; \mathbb{P}(\tilde{Y} = K|X)]$ and $\mathbf{P}$ is the one for $\mathbb{P}(Y = k|X)$.

For the second order information

$$\mathbb{P}(\tilde{Y}_1 = i, \tilde{Y}_2 = j|X) = \sum_{k \in [K]} \mathbb{P}(Y = k|X) \cdot \mathbb{P}(\tilde{Y}_1 = i|Y = k, X) \cdot \mathbb{P}(\tilde{Y}_2 = j|Y = k, X)$$

$$= \sum_{k \in [K]} \mathbb{P}(Y = k|X) \cdot T_{ki}(X) \cdot T_{kj}(X)$$

Enumerating pairs of $(i, j)$ we have $K^2$ equations, written in matrix form as:

$$C = (T(X))^\top \cdot \Lambda \cdot T(X)$$

where in above $C$ is a $K \times K$ matrix with the $(i, j)$-th entry being $\mathbb{P}(\tilde{Y}_1 = i, \tilde{Y}_2 = j|X)$; $\Lambda$ is a diagonal matrix with $\Lambda_{ii} = \mathbb{P}(Y = k|X)$.

Notice that

$$\sum_j \mathbb{P}(\tilde{Y}_1 = i, \tilde{Y}_2 = j|X) = \mathbb{P}(\tilde{Y}_1 = i)$$

and

$$\sum_j \sum_{k \in [K]} \mathbb{P}(Y = k|X) \cdot T_{ki}(X) \cdot T_{kj}(X) = \sum_{k \in [K]} \mathbb{P}(Y = k|X) \cdot T_{ki}(X)$$

we know that for every $K$ equations from the second order information, there is at least one redundant equation. That is to conclude that we have at most $K + K^2 - K = K^2$ independent equations. Nonetheless, we have $K(\mathbb{P}(Y = k|X)) + K^2(T(X)) = K^2 + K$ unknown variables. So the equations are under-determined. Therefore we conclude for the general $K$, there exists cases two labels will not define a unique solution. For instance, for $K = 3$, we can easily find the following two sets of parameter settings will return us the same observed distribution for two labels:

Parameter setting 1:

$$[\mathbb{P}(Y = 1|X), \mathbb{P}(Y = 2|X), \mathbb{P}(Y = 3|X)] = [0.35, 0.35, 0.3], T(X) = \begin{bmatrix} 0.6 & 0.2 & 0.2 \\ 0.175 & 0.65 & 0.175 \\ 0.15 & 0.15 & 0.7 \end{bmatrix}$$

Parameter setting 2:

$$[\mathbb{P}(Y = 1|X), \mathbb{P}(Y = 2|X), \mathbb{P}(Y = 3|X)] = [0.31, 0.34, 0.35], T(X) = \begin{bmatrix} 0.65 & 0.175 & 0.175 \\ 0.175 & 0.65 & 0.175 \\ 0.175 & 0.175 & 0.65 \end{bmatrix}$$

Similar examples can be obtained by searching through the solutions space of the equations. □

## PROOF FOR THEOREM 6

*Proof.* In the unstructured model, we first show that, with a large $N$, with high probability, each $X$'s will present at least 3 times. Denote by $N_X$ the number of times $X$ appears in the dataset. Then

$$N_X := \sum_{i=1}^N 1[X_i = X], \ \mathbb{E}[N_X] = \frac{q_X}{\sum_{X \in \mathcal{X}} q_X} N \tag{18}$$

When $N$ is large enough such that $N > \frac{4 \sum_{X \in \mathcal{X}} q_X}{\min_X q_X}$, we have $\mathbb{E}[N_X] > 4$. Then using Hoeffding inequality we have

$$\mathbb{P}(N_X \le 3) \le exp(-2N).$$

Using union bound (across $N$ samples), it implies that with probability at least $1 - N exp(-2N)$, $N_X \geq 3, \forall X$:

$$\mathbb{P}(N_X > 3, \forall X) = 1 - \mathbb{P}(N_X \leq 3, \exists X) \leq 1 - N exp(-2N) \tag{19}$$

This further implies that with probability at least $1 - N exp(-2N)$, we have $X_1 = X_2 = X$ for each $X$: Their distance is 0, clearly falling below the closeness threshold $\epsilon$. Therefore they will share the same true label.

Note that we are not imagining the exact same data appearing three times, but rather that three different data that happen to have the same pattern $X$ that appeared three times ([2]). For instance, these three $X$s can correspond to three independent users trying to apply for a credit card and ending up having the same application profiles (e.g., age, salary range, education level etc); it can also be three similar cat images ended up with the same encoding of the features. □

## PROOF FOR THEOREM 7

*Proof.* The $d^*$ features and the noisy label $\tilde{Y}$ jointly give us $d^* + 1$ independent observations. Denote by $K_G^* = \max_{X \in G} K_X$. In Kruskal's setup, $Y \in \{1, 2, ..., K_G^*\}$ will then correspond. to the unobserved hidden variable $Z$. If the noisy label is informative we know that $\mathsf{Kr}(T(X)) = K_G^* \leq K$. Then checking Kruskal's condition we have:

$$\mathsf{Kr}(T(X)) + \sum_{i=1}^{d^*} \mathsf{Kr}(M_i) \geq K_G^* + 2 \cdot d^* \geq K_G^* + K_G^* + d^* = 2K_G^* + d^* + 1 - 1$$

Calling Theorem 4, we establish the identifiability. □

## PROOF FOR THEOREM 8

*Proof.* By definition

$$||T_1(X) - T^*(X)||_F = \sqrt{\sum_i \sum_j (T_1[i,j] - T[i,j])^2} \tag{20}$$

Easy to show that

$$||T_1(X) - T^*(X)||_F + ||T_2(X) - T^*(X)||_F$$
$$= \sqrt{\sum_i \sum_j (T_1[i,j] - T[i,j])^2} + \sqrt{\sum_i \sum_j (T_2[i,j] - T[i,j])^2}$$
$$= \sqrt{\left( \sqrt{\sum_i \sum_j (T_1[i,j] - T[i,j])^2} + \sqrt{\sum_i \sum_j (T_2[i,j] - T[i,j])^2} \right)^2}$$
$$\geq \sqrt{\sum_i \sum_j \left( (T_1[i,j] - T[i,j])^2 + (T_2[i,j] - T[i,j])^2 \right)}$$

(Dropping the cross-product term which is positive)

Then we prove that

$$
\begin{aligned}
&||T_1(X) - T^*(X)||_F + ||T_2(X) - T^*(X)||_F \\
&\geq \sqrt{\sum_i \sum_j \left( T_1[i,j] - \frac{T_1[i,j] + T_2[i,j]}{2} \right)^2 + \left( T_2[i,j] - \frac{T_1[i,j] + T_2[i,j]}{2} \right)^2}
\end{aligned}
$$

(minimum distance is at half)

$$
\begin{aligned}
&= \sqrt{\sum_i \sum_j 2 \left( \frac{T_1[i,j] - T_2[i,j]}{2} \right)^2} \\
&= \frac{1}{\sqrt{2}} \sqrt{\sum_i \sum_j (T_1[i,j] - T_2[i,j])^2} \\
&= \frac{1}{\sqrt{2}} ||T_1(X) - T_2(X)||_F
\end{aligned}
$$

$\square$

## PROOF FOR THEOREM 9

*Proof.* The proof is straightforward by checking Kruskal's identifiability condition:

$$
\mathsf{Kr}(T(X)) + \sum_{i=1}^{d^*} \mathsf{Kr}(M_i) \geq 1 + 2 \cdot d^* \geq 1 + 2|G|K - 1 + d^* = 2|G| \cdot K + d^* + 1 - 1
$$

Note $|G| \cdot K$ is the size of space for the unobserved variable ($GY$ renumbered as $\{1, 2, ..., |G|K\}$). $\square$

## B  GENERIC IDENTIFIABILITY

We provide a bit more detail for the discussion on generic identifiability left in Section 5.2.

**Theorem 10.** *With a single informative noisy label, $T(X)$ is generically identifiable for each group $g \in G$ if the number of disentangled features $d^*$ satisfies that $d^* \geq \lceil \log_2 \frac{2K_G^* + 1}{2} \rceil$, and $\tau_i \geq 2$.*

*Proof.* We first reproduce a relevant theorem in (Allman et al., 2009):

**Theorem 11.** *(Allman et al., 2009) When $p = 3$ (3 independently observations), the model parameters are generically identifiable, up to label permutation, if*

$$
\min(K_G^*, \kappa_1) + \min(K_G^*, \kappa_2) + \min(K_G^*, \kappa_3) \geq 2K_G^* + 2 \tag{21}
$$

Based on the above theorem we have the following identifiability result:

Grouping $d^*$ features evenly into two groups, each corresponding to a meta variable/feature:

$$
R_1^* = \prod_{i=1}^{d_1^*} R_i, \; X_2^* = \prod_{j=d_1^*+1}^{d^*} R_j
$$

Denote feature dimensions of each group as $d_1^*, d_2^*$:

$$
\tau_1^* = \prod_{i=1}^{d_1^*} \geq 2^{d_1^*} \geq 2^{\lceil \log_2 \frac{2K_G^*+1}{2} \rceil} \geq \frac{2K_G^* + 1}{2} \tag{22}
$$

Similarly $\tau_2^* \geq \frac{K_X + 2}{2}$. Denote by $M_1^*, M_2^*$ the two observation matrices for the grouped variables

$$
M_i^*[j,k] = \mathbb{P}(R_i^* = \mathcal{R}_i^*[k]|Y = j), \; i = 1, 2.
$$

Then:

$$
\mathsf{Kr}(T(X)) + \mathsf{Kr}(M_1^*) + \mathsf{Kr}(M_2^*) \geq 1 + 2\frac{2K_G^* + 1}{2} = 2K_G^* + 2,
$$

which again satisfied the identifiability condition specified in Theorem 4. $\square$

---

**Algorithm 1** Key Steps of HOC

---

1: **Input:** Noisy dataset: $\widetilde{D} = \{(x_n, \tilde{y}_n)\}_{n \in [N]}$, with disentangled features.
   *//Find 2-NN using a similarity function $\mathsf{Sim}(x, x')$.*
2: With $1 - \mathsf{Sim}(x, x')$ as the distance metric:
   $\qquad \{(\tilde{y}_n, \tilde{y}_{n_1}, \tilde{y}_{n_2}), \forall n\} \leftarrow \mathtt{Get2NN}(\widetilde{D});$
   *//Count first-, second, and third-order consensus patterns:*
3: $(\hat{c}^{[1]}, \hat{c}^{[2]}, \hat{c}^{[3]}) \leftarrow \mathtt{CountFreq}(\{(\tilde{y}_n, \tilde{y}_{n_1}, \tilde{y}_{n_2}), \forall n\})$
   *//Solve equations:*
4: Find $T$ such that match the counts $(\hat{c}^{[1]}, \hat{c}^{[2]}, \hat{c}^{[3]})$.

---

## C   MORE EXPERIMENTS

In this section, we elaborate the detailed experiment setting and perform more experiments *w.r.t.* disentangled features.

### C.1   EXPERIMENT SETTING FOR TABLE 1

**Label Noise Generation** The label noise of each instance is characterized by $T_{ij}(X) = \mathbb{P}(\widetilde{Y} = j | X, Y = i)$. In this paper, we consider two types of label noise: asymmetric label noise (Han et al., 2018; Wei et al., 2020) and instance-dependent label noise (Cheng et al., 2021a; Zhu et al., 2021b). For asymmetric label noise, $T(X) \equiv T$, each clean label is randomly flipped to its adjacent label w.p. $\epsilon$, where $\epsilon$ is the noise rate, i.e., $T_{ii} = 1 - \epsilon$, $T_{ii} + T_{i,(i+1)_K} = 1$, $(i+1)_K := i \mod K + 1$. For instance-dependent label noise, the generation of noisy labels also depends on the features. We follow CORES (Cheng et al., 2021a) to generate instance-dependent label noise. The generation process is detailed in Algorithm 2. With these definitions, *asymm./inst.* $\epsilon$ in Table 1 denotes asymmetric/instance-dependent label noise with noise rate $\epsilon$.

**Model pre-training**. The network structures of all the three encoders in Table 1 are ResNet50 (He et al., 2016). Note that the encoders which generate features to estimate transition matrix can be pre-trained on different dataset. For example, HOC (Zhu et al., 2021c) utilizes ImageNet pre-trained encoders to generate features for CIFAR. Thus, following the pipeline of disentangled feature generation (Wang et al., 2021c) , we pre-train all the three encoders on CIFAR100 dataset and generate feature for CIFAR10 to estimate transition matrix. The first encoder is trained under 0.1 symmetric label noise rate to simulate the weakly-supervised features while the second and third encoder is trained via self-supervised learning (SSL). Recall the goal of SSL is to learn a good representation without accessing labels. In this paper, we adopt SimCLR (Chen et al., 2020) and IPIRM (Wang et al., 2021c) to perform SSL pre-training. SimCLR, as a representative work on SSL literature, learns a good represention based on InfoNCE loss (Van den Oord et al., 2018). However, it is shown that the features learned by SimCLR are only *partly* disentangled on some simple augmentation features such as rotation and colorization (Wang et al., 2021c). Thus IPIRM proposes a learning algorithm that embeds InfoNCE loss into IRM (Invariant Risk Minimization) framework (Arjovsky et al., 2019) to learn *fully* disentangled features. We train SimCLR model and IPIRM model by referring official codebase of IPIRM [4]. The pre-trained models, as well as evaluation code are all released in the supplementary material.

**Key steps of HOC**

**Estimation error of Transition matrix**. After training these three encoders, we fix the encoder and generate features from raw samples to estimate the noise transition matrix using Global HOC estimator (Zhu et al., 2021c). The hyper-parameters for estimating transition matrix are consistent with official implementation of HOC [5]: optimizer: Adam, learning rate: 0.1, number of iterations: 1500. After training, we evaluate the performance via absolute estimation error defined below:

$$\text{err} = \frac{\sum_{i=1}^{K} \sum_{j=1}^{K} |\hat{T}_{i,j} - T_{i,j}|}{K^2} * 100,$$

---

[4]https://github.com/Wangt-CN/IP-IRM
[5]https://github.com/UCSC-REAL/HOC

---

**Algorithm 2** Instance-Dependent Label Noise Generation

---

**Input:**

    1: Clean examples $(x_n, y_n)_{n=1}^N$; Noise rate: $\varepsilon$; Size of feature: $1 \times S$; Number of classes: $K$.

**Iteration:**

    2: Sample instance flip rates $q_n$ from the truncated normal distribution $\mathcal{N}(\varepsilon, 0.1^2, [0, 1])$;

    3: Sample $W \in \mathcal{R}^{S \times K}$ from the standard normal distribution $\mathcal{N}(0, 1^2)$;

    **for** $n = 1$ to $N$ **do**

    4:    $p = x_n \cdot W$    // Generate instance dependent flip rates. The size of $p$ is $1 \times K$.

    5:    $p_{y_n} = -\infty$    // Only consider entries different from the true label

    6:    $p = q_n \cdot \text{softmax}(p)$    // Let $q_n$ be the probability of getting a wrong label

    7:    $p_{y_n} = 1 - q_n$    // Keep clean w.p. $1 - q_n$

    8:    Randomly choose a label from the label space as noisy label $\tilde{y}_n$ according to $p$;

    **end for**

**Output:**

    9: Noisy examples $(x_i, \tilde{y}_n)_{n=1}^N$.

---

Table 2: Comparison of test accuracy on CIFAR10 by using the estimated transition matrix.

| Methods | inst. 0.3 | inst. 0.4 | inst. 0.5 | inst. 0.6 |
|---|---|---|---|---|
| FW (SimCLR) | 66.61 | 65.82 | 64.51 | 62.81 |
| FW (IPIRM) | 73.24 | 72.54 | 71.33 | 69.42 |

Table 3: Comparison of test accuracy on CIFAR100 by using different DNN initialization.

| Methods | inst. 0.3 | inst. 0.4 | inst. 0.5 | inst. 0.6 |
|---|---|---|---|---|
| CE (random init) | 43.47 | 35.17 | 27.07 | 18.25 |
| CE (SimCLR init) | 58.95 | 49.7 | 36.87 | 25.07 |
| CE (IPIRM init) | 64.92 | 56.18 | 43.75 | 30.36 |

where $\hat{T}$ is the estimated noise transition matrix, $T$ is the real noise-transition matrix, $K$ is the number of classes in the dataset.

## C.2   Training performance using estimated transition matrix

We can further use the estimated transition matrix to perform forward loss correction (FW) (Patrini et al., 2017b). Table 2 records the performance of FW by using the estimated transition matrix of SimCLR and IPIRM. The hyper-parameters for all the experiments in Table 2 are the same: optimizer: SGD, training epochs: 100, learning rate: 0.1 for first 50 epochs and 0.01 for last 50 epochs, batch-size: 256. From the results, we can observe that the test accuracy increases as features become more disentangled.

## C.3   Initializing DNN using disentangled features

Except for estimating transition matrix, we can directly use disentangled features to perform training on noisy dataset. Table 3 shows the effect of using disentangled features as DNN initialization on CIFAR100. The hyper-parameters for all the experiments in Table 3 are consistent with Table 2. From the results, We can observe that even with vanilla Cross Entropy loss, the disentangled features are still beneficial to the performance.

Table 4: Comparison of test accuracy on CIFAR10 by using different transition matrix.

|               | CE    | FW with $T_1$ | FW with $T_2$ | FW with $T_3$ |
| ------------- | ----- | ------------- | ------------- | ------------- |
| Test accuracy | 79.34 | 82.62         | 81.65         | 78.13         |

## C.4 VERIFYING THE IMPORTANCE OF CHARACTERIZING THE IDENTIFIABILITY OF THE LABEL NOISE TRANSITION MATRIX

### C.4.1 CIFAR10 EXPERIMENT

Our first experiment is to show that when estimated transition matrices is far from the ground-truth matrix, it may make model perform worse even compared to the baseline (vanilla training with Cross Entropy).

**Experiment setting:** The training framework with transition matrix is followed from FW (Patrini et al., 2017b). The dataset is CIFAR10 and the network structure is ResNet34. The hyper-parameters are as follows: batchsize (64), learning rate (0.1 for first 50 epochs and 0.01 for last 50 epochs), optimizer (SGD). For a randomly selected set of instances ($0\%$ of the population), we generate noisy labels using the following transition matrix:

$$T = \mathbb{P}(\widetilde{Y}|Y,X) = \begin{bmatrix} \mathbf{0.9} & 0.011 & 0.011 & 0.011 & 0.011 & 0.011 & 0.011 & 0.011 & 0.011 & 0.011 \\ 0.019 & \mathbf{0.82} & 0.019 & 0.019 & 0.019 & 0.019 & 0.019 & 0.019 & 0.019 & 0.019 \\ 0.028 & 0.028 & \mathbf{0.74} & 0.028 & 0.028 & 0.028 & 0.028 & 0.028 & 0.028 & 0.028 \\ 0.037 & 0.037 & 0.037 & \mathbf{0.66} & 0.037 & 0.037 & 0.037 & 0.037 & 0.037 & 0.037 \\ 0.045 & 0.045 & 0.045 & 0.045 & \mathbf{0.58} & 0.045 & 0.045 & 0.045 & 0.045 & 0.045 \\ 0.054 & 0.054 & 0.054 & 0.054 & 0.054 & \mathbf{0.51} & 0.054 & 0.054 & 0.054 & 0.054 \\ 0.063 & 0.063 & 0.063 & 0.063 & 0.063 & 0.063 & \mathbf{0.43} & 0.063 & 0.063 & 0.063 \\ 0.071 & 0.071 & 0.071 & 0.071 & 0.071 & 0.071 & 0.071 & \mathbf{0.35} & 0.071 & 0.071 \\ 0.08 & 0.08 & 0.08 & 0.08 & 0.08 & 0.08 & 0.08 & 0.08 & \mathbf{0.27} & 0.08 \\ 0.088 & 0.088 & 0.088 & 0.088 & 0.088 & 0.088 & 0.088 & 0.088 & 0.088 & \mathbf{0.2} \end{bmatrix}$$

The above transition matrix $T$ is uniform off-diagonal with diagonals evenly spaced over $[0.9, 0.2]$, which is the ground-truth transition matrix in our setting. The remaining unselected instances will enjoy a $T \equiv 0$.

We perform experiments using the following three uniform off-diagonal transition matrix with forward loss correction (Patrini et al., 2017b):

- $T_1$ with diagonals evenly spaced over $[0.9, 0.2]$

- $T_2$ with all diagonals $0.4$.

- $T_3$ with diagonals evenly spaced over $[0.2, 0.9]$

where $T_1$ is the ground-truth transition matrix while $T_3$ is far from the ground-truth. The results are listed in Table 4.

It can be observed that when using $T_3$, the performance is even worse than vanilla training with Cross Entropy, suggesting the importance of identifying and estimating the noise transition matrix.

### C.4.2 GAUSSIAN EXPERIMENT

Our second experiment is to show that in some settings, the transition matrix is hard to estimate correctly, which suggests the importance of identifiability. Consider a simple setting for binary classification and a set of instances generated according to the following setups:

- $X \sim \mathcal{N}(0, 3)$ where $\mathcal{N}$ denotes Gaussian distribution with mean 0 and variance 3.

- $\mathbb{P}(Y = 1|X) = \text{sigmoid}(X) = \frac{1}{1+e^{-X}}$

Table 5: Comparison of test accuracy for CE and FW.

|  | CE | FW with estimated transition matrix |
|---|---|---|
| Test accuracy | 83.22 | 83.31 |

We generate $X$ and $Y$ following the above procedure and define the ground-truth transition matrix $T = \mathbb{P}(\widetilde{Y}|Y, X) = \begin{bmatrix} \mathbf{0.9} & 0.1 \\ 0.2 & \mathbf{0.8} \end{bmatrix}$ for generating $\widetilde{Y}$ from $Y$. Our goal is to examine whether we can estimate the correct transition matrix using $(X, \widetilde{Y})$.

**Experiment setting:** The training framework for estimating transition matrix is followed from FW (Patrini et al., 2017b). We randomly sample 5000 $(x, y)$ pairs from the data generating procedure and using $T = \mathbb{P}(\widetilde{Y}|Y) = \begin{bmatrix} \mathbf{0.9} & 0.1 \\ 0.2 & \mathbf{0.8} \end{bmatrix}$ to generate $\widetilde{Y}$ from $Y$. The network structure is a simple FCN (fully connected network tructure) with one hidden layer (10 nodes) and ReLU activation. The hyper-parameters are as follows: learning rate (0.01 for 100 epochs ), optimizer (SGD). We perform the experiments with 30 runs and record the average performance in Table 5.

From Table 5, we can see that FW has very little gain compared to vanilla Cross Entropy training. We then calculate the average estimated transition matrix:

$$T_{estimated} = \begin{bmatrix} \mathbf{0.983} & 0.017 \\ 0.008 & \mathbf{0.992} \end{bmatrix}$$

We find that $T_{estimated}$ is nearly as the same as the identity matrix, suggesting that in this setting, FW is hard to estimate noise transition matrix correctly and contributes less to the performance.

$$[\mathbb{P}(Y = 1|X), \mathbb{P}(Y = 2|X), \mathbb{P}(Y = 3|X)] = [0.35, 0.35, 0.3], T(X) = \begin{bmatrix} 0.6 & 0.2 & 0.2 \\ 0.175 & 0.65 & 0.175 \\ 0.15 & 0.15 & 0.7 \end{bmatrix}$$

