# OpenReview forum: "Identifiability of Label Noise Transition Matrix "
_ICLR.cc/2023/Conference — Submitted to ICLR 2023_

### Official Review · Reviewer_14GL · 2022-10-23

**Confidence:** 3
**Correctness:** 2
**Technical Novelty And Significance:** 3
**Empirical Novelty And Significance:** 3
**Recommendation:** 3

**Clarity, Quality, Novelty And Reproducibility:**

- Clarity & quality: some parts of the paper are unclear (please see the weakness part above for more details).
- Novelty: It is an interesting view to understand the identifiability of the noise transition matrix by using the Kruskal's identifiability results.
- Reproducibility: the paper has provided the code and complete proofs for the theorems.


**Strength And Weaknesses:**

### Strength
- The paper provides a sufficient and necessary condition for learning with instance-dependent noise. Although the main theorem (Theorem 5) seems a direct application of Kruskal's identifiability results, to me, it is still an interesting and novel view for understanding the identifiability of the label noise learning problem.
- Experiments are conducted to show that the disentangled features help identify the transition matrix.

### Weaknesses
My main concern about the paper is the significance of the theorems presented in Section 5 and the clarity of the presentation.
- about the significance: In Section 5, the authors attempt to show that the requirement on three independent noisy labels can be relaxed by considering the smoothness of clusterability of the features. But, I am confused by some parts of the statement, and it is unclear how the proposed theorems could relate to the goal of relaxing multiple noisy labels.
	- About Definition 5: Definition 5 introduces the 2-NN clustrability. It is unclear to me how the clustrability could help to reduce the requirement of multiple noisy labels. It seems that we still need two additional conditions to ensure the identifiability: (1) the 2-nearest neighbors (NN) of $X$ shares the same transition matrix with $X$. (2) the noisy label of X and its 2-NN should be independent. It is unclear how easy it is to meet these conditions in practice. It seems that these conditions are somewhat equal to the three independent noisy label requirements in the proposed example (see below).
	- About the example in Theorem 6: the authors prove Theorem 6 by showing that the same instance $X$ will appear three times, and thus $X$ is the 2-NN of itself. It is unclear to me how the condition on the independence of the noisy labels could be satisfied in such a case. If the independent condition is true, it seems that we still need three independent nosy labels for the same $X$.

	- About Theorem 8 and Theorem 9: At first glance, the theorem shows that we only need 1 noisy label to ensure the identifiability of $T(X)$ when the representations of X are disentangled and informative. But it is unclear to me what is role of the disentangled feature plays in the theorems. It seems that, by the Kruskal's identifiability results, one can directly show $P(Y\vert X)$ is identifiable with the disentangled and informative representations only (please correct me if I am wrong). In such a case, there is no need to identify the noise transition matrix, which makes the requirement for disentangled and informative representations seem very strong.
- about the clarity of presentation:
	- This paper has introduced many notations, which makes it hard to follow. To me, some of them are somewhat redundant. For example, the notation of $P_\theta$ is seldomly used in the rest parts.
	- Section 3.1 has spent much space introducing the MPE problem. It is unclear how the MPE problem is related to the main focus of the paper presented in Section 4 and Section 5. I think it would be better for the authors to make this point clear.

### Minor points:
- It would be better to use \citep instead of \citet, if the cited paper does not appear as a component of the sentence.
- Page 6: the third line of the proof sketch, there is an additional "." after $Y\in\\{1,2,\dots,K\\}$.
- Page 6, first sentence in Section 5: focuses--> focus
- Definition 5: since $Y$ is a random variable,I think it would be better to write the condition as $P(Y \vert X) = P(Y_1 \vert X) =P(Y_2 \vert X)$ to avoid potential misunderstanding.
- Figure 2:"Grey"--> "Grey color"


**Summary Of The Paper:**

This paper studies the identifiability of the label noise transition matrix $T(X)$ for instance-dependent noises. By applying Kruskal's identifiability results to the label noise case, the authors have shown that three independent noisy labels for each instance are sufficient and necessary for identifying the transition matrix. Since three independent noisy labels could be a somewhat strong requirement, the paper has also discussed the identifiability of T(X) in the cases when $P(Y\vert X)$ enjoys clustering properties or the representations of the feature are disentangled and informative.


**Summary Of The Review:**

This paper has provided a sufficient and necessary condition for the identifiability of the noise transition matrix (in Section 4). Although the result seems a direct application of the Kruskal's identifiability results, it is still novel and interesting for me from the view of learning with noisy labels. My main concern about the paper is the significance of the theorems and the clarity of some statements. Overall, I think the flaws overweight the merits of this paper, and it falls into the borderline cases.

===post-rebuttal===
After reading the author's feedback and other reviews, I do find this paper has some interesting parts. But, as also mentioned by other reviewers, the clarity of this paper should be further improved, especially on the definition and the motivations of examples in Section 4. To me, the paper still needs a non-trivial revision before publication. Therefore, I tend to reject this paper.

---

> ### Author Response · Authors · 2022-11-13
> **Response to Reviewer 14GL**
>
> Thank you for your suggestion and comments. We have now added more clarifications to our draft and explain below. We hope this response will help remove some of the uncertainties.
>
> **Additional requirements**
>
> The reviewer is correct that the smoothness of $X$ also requires that  $(X, X_1, X_2)$ will have the same $T(X)$. This was indeed required in the original paper [1] that we cited and intended to analyze. We have now clarified this in Definition 5. Thank you for the catch.
>
> Theorem 6: in this model, we are not imagining the exact same data appearing three times, but rather that three different data that happen to have the same pattern $X$ that appeared three times ([2]). For instance, these three $ X$s can correspond to three independent users trying to apply for a credit card and ending up having the same application profiles (e.g., age, salary range, education level etc); it can also be three similar cat images ended up with the same encoding of the features. We added a discussion in the proof.
>
> With three independent tasks, the independence assumption is a commonly adopted one in the learning with noisy label community and admittedly has been a simplification to enable analysis.
>
> **Disentangled features:**
>
> Disentangled features serve as the observation for the unobserved variable $Y$. The literature of identifiability builds on the idea that with more conditionally independent observed variables, the problem will become more identifiable. We hope to explain why a single noisy label often suffices in practice in learning with noisy labels, and one promising explanation we found is that the $X$ also encodes observable and highly correlating information for the estimation task.
>
> The disentangled features do not immediately imply the estimation of $P(Y|X)$. The hidden structure model can be extended to capture $P(X|Y)$ (from the hidden variable to the observed variable), that is, given a true label being a certain class, what is the generative model for $X$. Our proved conditions don’t imply the identifiability challenge in estimating $P(X|Y)$ - this is primarily because of the high parameter space of the information structure. It would either imply that under our current proved conditions, $P(X|Y)$ is not guaranteed to be estimated, or that we would need a much higher number of disentangled features to perform the study.
>
> **Some of the notations**
>
> We agree the notations are heavy. The authors had a hard debate too and in the end, made a sacrifice to slightly overload the notation to follow the norm in the literature. We will explore to think about ways to simply.
>
> For the particular example, in the identifiability literature, the complete parameter space is defined by the priors of the hidden variables (in our case the true label $Y$), and the generation structure between the hidden label $Y$ and the observed variable (in our case the noisy labels). Therefore, for a complete and rigorous description of the identifiability problem, we introduced $\theta$ for the parameter space
>
> **MPE**
>
> The discussion of MPE is mainly to draw connections with the existing attempts towards establishing identifiability in the literature on learning with label noise, as well as to highlight the limitations in the current results and to introduce ours in the next following sections (e.g., focusing on binary classification problem, class-dependent label noise model etc). There is no other deeper connection.
>
> Others:
>
> Thank you for the catches. We have fixed them!
>
> **Def 5:** we do seem to require the realized true label to be the same across the three instances, instead of the distributions of the three random variables. So this is indeed a stronger requirement.
>
>
> [1] Clusterability as an Alternative to Anchor Points When Learning with Noisy Labels. ICML 2021.
>
> [2] Understanding Instance-Level Label Noise: Disparate Impacts and Treatments. ICML 2021.

---

### Official Review · Reviewer_4T3g · 2022-10-24

**Confidence:** 4
**Correctness:** 3
**Technical Novelty And Significance:** 3
**Empirical Novelty And Significance:** 3
**Recommendation:** 5

**Clarity, Quality, Novelty And Reproducibility:**

-Clarity: the paper is well structured and flows smoothly, but some unclear points regarding the most important identifiability definition of the transition matrix T should be clarified.

-Quality: the paper studies the identifiability conditions of the transition matrix in instance-dependent label noise learning and draws connections to many existing works on this problem, my biggest concern on this is the proof for necessity in Theorem 5. The paper would be stronger if this can be improved.

-Novelty: the paper draws interesting connections between the identifiability of the transition matrix in label noise learning and Kruskal’s identifiability results which is novel to me.

-Reproducibility: good.

**Strength And Weaknesses:**

Pros:

-Existing results on the identifiability of noise transition matrix mainly focus on the class-dependent instance-independent label noise. This paper considers this identifiability problem in the instance-dependent noise setting. They show that given three noisy labels the identifiability of an instance-dependent noise transition matrix can be guaranteed.

Cons:

-**The most important definition is unclear.** In the definition of identifiability of transition matrix $T(x)$, only $\theta(x)$ appears, it is not clear where the hidden variables $Y$ (unobserved) and $\tilde{Y}$ (observed) are. It also seems that the notation $\theta(X)$ is redundant, definition 2 can be explained directly using $P_{T(X)}$, as in $\theta(X)=(T(X), P(Y\mid X))$ the prior $P(Y\mid X)$ is not really a parameter (adding this makes me very confused at first glance). What is the statistical model in definition 2, given the current definition I think it should be $P_{\theta(x)}$, can we interpret it as a probability distribution over a probability? Then should definition 2 be put as $P_{\theta(x)}$ is identifiable if $\theta(x)\neq\theta’(x)$, $P_{\theta(x)}\neq P_{\theta’(x)}$? Also, why the label permutation appears in definition 2? Due to this condition, the notion is not identifiability to me.

-**The most important theorem is not rigorously proved.** In the identifiability condition (Theorem 5) proof for necessity, the authors **only prove the binary label noise case and directly drop the noise dependency on the instance**, but the paper claims a multi-class setting and the identifiability condition is both necessary and sufficient. At least this should be made clear in the main paper!! A natural question is can this necessity proof be extendable to the multi-class instance-dependent noise case?


**Summary Of The Paper:**

-This paper studies the identifiability conditions of the noise transition matrix in instance-dependent label noise learning. The authors draw interesting connections between this problem with the Kruskal’s identifiability results and provide conditions to guarantee the identifiability of the instance-dependent noise transition matrix. They also show some existing works can be explained by their findings and some empirical results demonstrating disentangled features help identify the noise transition matrix.

**Summary Of The Review:**

-The paper studies the challenging problem of identifying transition matrix in instance-dependent label noise setting and provides interesting results on identifiability conditions based on Kruskal’s identifiability results, but the unclear parts of the identifiability definition should be clarified and more theoretical analysis is needed.

---

> ### Author Response · Authors · 2022-11-13
> **Response to Reviewer 4T3g Part I**
>
> Thank you for your suggestion and comments. We agree with some of the raised confusion and we’d like to use the opportunity to provide further clarifications. We have also revised the draft and added new analysis. Hopefully, this response will help remove some of the uncertainties.
>
> **Definition of parameter space**
>
> In the definition of identifiability, $\theta(X):=\{T(X),\mathbb P(Y|X)\}$, defined 4 lines above Definition 2. Unobserved $Y$ and observed $\tilde Y$ are defined in the first paragraph of Section 2.
>
> In the identifiability literature, the complete parameter space is defined by the priors of the hidden variables (in our case the true label $Y$), and the generation structure between the hidden label Y and the observed variable (in our case the noisy labels). Therefore, for a complete description of the identifiability problem the parameter space has to be defined on the combined parameter space. For instance, even with the same $T(X)$, different prior of $P(Y|X)$ (e.g. for binary case a [0.8, 0.2] prior vs a [0.7, 0.3] prior) would return very different observed distributions of the noisy labels. On the other hand, combined with different priors and $T(X)$, one could have the same observed distribution - see the example in our response below for label permutation.
>
> In the hidden graphical model literature, $P(Y|X)$ is indeed viewed as the hidden parameter that we don’t really observe and therefore has to be jointly inferred. In fact, some of the successes in the literature (e.g., [1]) in estimating the noise transition matrix also often return the estimation of the true prior.
>
> In definition 2, $P_{\theta(X)}$ is the probability density function over the observed variables, given the parameter \theta(X). It is not a distribution over a probability but rather a distribution over a set of random variables. As discussed in the introduction, the reason we don’t fully spell them out is at the moment, we haven’t specified the full set of observed variables. Later in the paper, we considered the case with one observed noisy label, three such labels, and disentangled features.
>
>
> **Label permutation**
>
> Label permutation is not on reshuffling the label information but describes the following phenomenon in the literature: in a counterfactual world, one can relabel the **label space** and manipulate the parameters in a certain way such that the distribution of the observed variables is identical. These are often referred to as mirror cases with each other, and without additional information, one typically doesn't expect to tell one case from another.
>
> For example, imagine if one could swap the numbering of label space, say 0->1 and 1->0 (for example, we build a binary classifier for distinguishing ‘cat’ from ‘dog’. Image in a counterfactual world, where the language labels ‘cat’ as ‘dog’ and ‘dog’ as ‘cat’), there always exists a counterfactual error rate/noise model ($T(X)$) on the swapped label space that leads to the same observed distribution of noisy labels.  This is an entire mirror situation that would be hard to detect.
>
> For instance, suppose for the original label space, the true priors are $P(Y=1) = 0.6, P(Y=0) = 0.4$, with error rate $e_1 = 0.2, e_0 = 0.2$.
> Then
>
> $
> P(\tilde Y = 1) = P(Y=1) \cdot P(\tilde Y = 1|Y=1) +P(Y=0)\cdot P(\tilde Y = 1|Y=0)$
>
> $  = P(Y=1) \cdot (1-e_1)+P(Y=0)\cdot e_0 = 0.6\cdot 0.8+0.4\cdot 0.2 = 0.56$
>
> Now after swapping the label space, label 0 is relabeled as 1, and label 1 is relabeled as 0. Denote this new label as $Y’$. Now we have  $P(Y' =1) = 0.4, P(Y'=0) = 0.6$. Now flipping the error rates to $e'_0:=1-e_1$ and $e'_1:=1-e_0$ would result in the same observed distribution. To see this:
> $P(Y'=1) = 0.4, P(Y'=0) = 0.6$, and,
>
> $P(\tilde Y = 1) = P(Y'=1)\cdot (1-e'_1)+P(Y'=0) \cdot e'_0$
>
> $= 0.4 \cdot 0.2+0.6 \cdot 0.8 = 0.56$
>
> That is the two sets of error rates, along with priors, will return us identical distributions of noisy labels.
>
> Typically the restriction of noise rate specified that the error rate has to be below 0.5, removing the second case above with  $e'_0 = e'_1 = 0.8$.

---

> > ### Author Response · Authors · 2022-11-13
> > **Response to Reviewer 4T3g Part II**
> >
> > **Theorem 5**
> >
> > The sufficiency proof in Theorem 5 is not customized to binary class and holds for multi-class as well. The binary class argument suffices for the necessary proof (that is there exists a case that two labels failed to identify the transition matrix) but also that it presents the idea more transparently, in the author’s opinion. We can construct a similar multi-class argument:
> > the core is to show that there can exist multiple matrices $T_1, T_2$ along with the priors $p_1, p_2$ such that $P(\tilde Y_1, \tilde Y_2|X)$ and $P(\tilde Y_1|X)$ would be identical.
> >
> > We have now added proof for the multi-class case (Appendix, pages 17-18, proof of Theorem 5). The basic idea is to show that the full information one can extract returns us an equation with at most $K^2$ independent equations but for $K^2+K$ unknown variables, therefore the equation is under-defined. Based on the equation characterization, one can easily search for counter-examples. For instance, for $K=3$, we present the following cases (allowing numeric errors):
> >
> > Parameter setting 1:
> >
> > $[P(Y=1|X), P(Y=2|X), P(Y=3|X)] = [0.35,0.35,0.3]$
> >
> > $T(X) =$
> >
> > $[[0.6,0.2,0.2],$
> >
> > $[0.175,0.65,0.175],$
> >
> > $[0.15,0.15,0.7]]$
> >
> > Parameter setting 2:
> > $[P(Y=1|X), P(Y=2|X), P(Y=3|X)] = [0.31,0.34,0.35]$
> >
> > $T(X) =$
> >
> > $[[0.65,0.175,0.175],$
> >
> > $[0.175,0.65,0.175],$
> >
> > $[0.175,0.175,0.65]]$
> >
> > It can be verified easily that the above two settings return identical distributions for the two noisy labels.
> >
> > The proof is indeed for instance dependent case since it is an argument for each single $X$ - we dropped the notation dependency for the ease of presentation. It is easy to extend the proof to more instance-dependent cases: Suppose $X$s are grouped into two groups, with each of them having one pair of the error rates specified in the proof (that lead to the same distribution of one and two labels). We will not be able to identify either one by observing only two labels.
> >
> >
> > [1] Clusterability as an Alternative to Anchor Points When Learning with Noisy Labels. ICML 2021.

---

> > > ### Comment · Reviewer_4T3g · 2022-11-23
> > > **Thanks for the response**
> > >
> > > I have read the response, but it's still very confusing about the definition of $\theta(X)$ and the identifiability of $T(X)$. Thanks for the example on different priors, which makes sense to me. However, $T(X)$ depends on $Y$ and $\tilde{Y}$, $P(Y|X)$ also depends on $Y$, $\theta(X)$ no longer depend on $Y$. Or do you assume specific parametric models; if so, it is not clear how the identifiability can be tackled in these unspecified parametric models then. I still think more clarifications and improvements are needed for the current work.

---

> > > > ### Author Response · Authors · 2022-11-24
> > > > **follow up**
> > > >
> > > > Please correct us we misunderstand the reviewer's question but in our setting we do not make any assumptions about $\theta(X)$ and for rigorousness and ease of presentation, $\theta(X)$ is merely an introduced notation to denote a combined parameter space of both $T(X)$ and $P(Y|X)$, nothing more or less.
> > > >
> > > > For a simple example, say we have two variables $Z_1, Z_2$ with dimension 1. We then use $Z$ to denote the combined space $[Z_1, Z_2]$ which is of dimension 2 - $\theta(X)$ has the exact same role as $Z$ here, simply denoting both sets of parameters $T(X)$ and $P(Y|X)$ in a compact way.
> > > >
> > > > Therefore if $T(X)$ and $P(Y|X)$ depend on $Y$, the combined parameter $\theta(X)$ depends on $Y$. Furthermore, the identifiability of $\theta(X)$ will also imply the identifiability of each of its components, namely $T(X)$ and $P(Y|X)$.
> > > >
> > > > Please let us know if there are other causes of confusion. Happy to clarify.

---

> > > > > ### Author Response · Authors · 2022-12-03
> > > > > **gentle follow up**
> > > > >
> > > > > Dear Reviewer 4T3g,
> > > > >
> > > > > Thank you for your comments and discussions. We are following up to see if there is anything that we could further clarify. Thank you.
> > > > >
> > > > > Best,
> > > > >
> > > > > Authors

---

### Official Review · Reviewer_GkBx · 2022-10-25

**Confidence:** 3
**Correctness:** 4
**Technical Novelty And Significance:** 3
**Empirical Novelty And Significance:** 3
**Recommendation:** 6

**Clarity, Quality, Novelty And Reproducibility:**

Most of the paper is clear and well-written. The paper is of good quality and gives novel results. Source code is provided and sufficient details are given for reproducibility.


**Strength And Weaknesses:**

Strengths:
Identifiability of T(X) at the instance level is an important and challenging problem. The paper's results tell that we would need three conditionally independent and informative noisy labels to identify it. The results have practical implications on various settings e.g. crowdsourcing etc. where multiple noisy labels are acquired and then aggregated to obtain ground truth labels. Some of their results verify the empirical success of existing approaches from the identifiability perspective.  They empirically show the possibility of learning disentangled features to help identify the noise transition matrix.

Questions/Weaknesses:
Are there any assumptions on the noise levels? In particular do the results in theorem 5 hold for arbitrary noise level? How does the cardinality(size) of label space play a role here -- looks like the problem will be harder as the label space grows? How do the results compare against prior works like Liu et al. (2020); Zhu et al. (2021c); Zhang et al. (2014).

**Summary Of The Paper:**

The paper studies identifiability of label noise transition matrix T(X), which plays a crucial role in learning with noisy labels. Most of the works assume access to it or rely on some methods to estimate it. Understanding when such a noise matrix is identifiable is an important aspect of the problem. The paper considers instance dependent noise transition matrices and show the necessity of multiple noisy labels in identifying the noise transition matrix at the instance level. Their analysis builds on top of Kruskal's identifiability results.  Their analysis also sheds light on the successes of SOTA methods and how additional assumptions helped them alleviate the requirement of multiple noisy labels. They also provide empirical evidence to show that disentangled features are helpful in the identification task.

**Summary Of The Review:**

The paper gives several insightful results on identifiability of label noise transition matrix and these results have practical implications. More clarity on the setup and results can provided to make the better.

---

> ### Author Response · Authors · 2022-11-13
> **Response to Reviewer GkBx**
>
> Thank you for your suggestion and comments. We now provide clarifications. Hopefully, this response will help remove some of the uncertainties.
>
> **Requirement of noise rate**
>
> There is no requirement for the noise rate - this is because the identifiability condition is defined under label permutation. So the typical mirror cases (e.g., error rate being 0.3 or 0.7) are considered equally identifiable. Typically the restriction of noise rate specified that the error rate has to be below 0.5, removing the second case above. But in our setting, these two cases are regarded as being equivalent.
>
> Consider the following example:  Consider in a counterfactual world, if one could swap the numbering of label space, say 0->1 and 1->0 (for example, we build a binary classifier for distinguishing ‘cat’ from ‘dog’. Image in a counterfactual world, where the language labels ‘cat’ as ‘dog’ and ‘dog’ as ‘cat’), there always exists a counterfactual error rate/noise model ($T(X)$) on the swapped label space that leads to the same observed distribution of noisy labels.  This is an entire mirror situation that would be hard to detect.
>
> For instance, suppose for the original label space, the true priors are $P(Y=1) = 0.6, P(Y=0) = 0.4$, with error rate $e_1 = 0.2, e_0 = 0.2$.
> Then
>
> $
> P(\tilde Y = 1) = P(Y=1) \cdot P(\tilde Y = 1|Y=1) +P(Y=0)\cdot P(\tilde Y = 1|Y=0)$
>
> $  = P(Y=1) \cdot (1-e_1)+P(Y=0)\cdot e_0 = 0.6\cdot 0.8+0.4\cdot 0.2 = 0.56$
>
> Now after swapping the label space, label 0 is relabeled as 1, and label 1 is relabeled as 0. Denote this new label as $Y’$. Now we have  $P(Y' =1) = 0.4, P(Y'=0) = 0.6$. Now flipping the error rates to $e'_0:=1-e_1$ and $e'_1:=1-e_0$ would result in the same observed distribution. To see this:
> $P(Y'=1) = 0.4, P(Y'=0) = 0.6$, and,
>
> $P(\tilde Y = 1) = P(Y'=1)\cdot (1-e'_1)+P(Y'=0) \cdot e'_0$
>
> $= 0.4 \cdot 0.2+0.6 \cdot 0.8 = 0.56$
>
> That is the two sets of error rates, along with priors, will return us identical distributions of noisy labels.
>
> The restriction of noise rate being less than 0.5 removes the second case above with  $e'_0 = e'_1 = 0.8$. But in our setting, these two cases are regarded as being equivalent.
>
>
> **Number of classes**
>
> For our results in Section 4, when the three conditionally independent labels are available, our results are not directly dependent on the number of classes. The intuition is as follows: the Kruskal rank of the noise transition matrix encodes the amount of information needed. When the label space grows, the identifiability condition (RHS of the condition equation) does become harder, but the observed information from the noisy labels is also becoming richer (encoded by Kruskal rank). But the reviewer is 100% correct about our results in Section 5. When there is only one noisy label, a higher dimension of the label space implies a higher requirement of the disentangled features.
>
>
> **Comparisons to prior works**
>
> We would like to emphasize that our study doesn’t propose an algorithm that relies on a particular collection of datasets. Rather, our identifiability results specify the required property of data in order to identify its hidden noise model $T(X)$. The results provide guidelines for when $T(X)$ would be identifiable or not for an $X$, where the conditions specified the requirements for the noisy labels that can be observed for $X$, and how informative $X$ itself is. In other words, our results helped explain why the listed prior works (Liu et al. (2020); Zhu et al. (2021c); Zhang et al. (2014)) are able to estimate the noise transitions correctly (e.g., three independent noisy labels). This inquiry is independent of the design of a specific algorithm and contributes parallelly to the prior works listed.
>
> When the conditions don’t satisfy, our results in fact provide guidelines for possible solutions to curate the data, including performing better representation disentangling, and soliciting more independent labels. We believe this contribution is new and complementary to the prior research as well.

---

### Official Review · Reviewer_G5Yr · 2022-10-28

**Confidence:** 3
**Correctness:** 3
**Technical Novelty And Significance:** 2
**Empirical Novelty And Significance:** 2
**Recommendation:** 5

**Clarity, Quality, Novelty And Reproducibility:**

Clarity: Poor
Quality: Poor
Novelty: Good
Reproducibility: Not sure

**Strength And Weaknesses:**

--Strength--
1. It characterizes the identifiability of the label noise transition matrix for the generic case at the instance level.

--Weakness--
1. The importance and necessity of studying the label noise transition matrix's identifiability are unclear. Additionally, why is it important to consider the instance-dependent label noise case?
2. This paper lacks experiments to verify the importance of characterizing the identifiability of the label noise transition matrix at the instance level.
3. The format of the cited paper is not proper and seems to lack brackets.
4. Although the authors find that three separate independent and identically distributed (i.i.d.) noisy labels (random variables) are both necessary and sufficient for instance-level identifiability, are these noise types sufficient to describe the real label noise?

**Summary Of The Paper:**

This paper characterizes the identifiability of the label noise transition matrix for the generic case at the instance level based on Kruskal’s identifiability results. The main contribution is that it finds three separate independent and identically distributed (i.i.d.) noisy labels (random variables) are both necessary and sufficient for instance-level identifiability and the disentangled features help with identifiability.

**Summary Of The Review:**

This paper characterizes the identifiability of the label noise transition matrix for the generic case at the instance level based on Kruskal’s identifiability results. However, this paper is not well written and some points are not well explained, especially the importance of studying the identifiability and the use of the proposed theorem in this paper. Therefore, this paper is marginally below the acceptance threshold.

---

> ### Author Response · Authors · 2022-11-13
> **Response to Reviewer G5Yr Part I**
>
> Thank you for your suggestion and comments. We now have added new experiments and clarifications. Hopefully, this response will help remove some of the uncertainties.
>
> **Why identifiability?**
>
> Identifying the noise transition matrix is a crucial task in the problem of learning with noisy labels. The literature misses a unified understanding of when the noise transition matrix is identifiable, particularly true for the recently surging discussions concerning instance-dependent noise. When a learning problem becomes unidentifiable, it remains unclear whether the underlying problem is learnable, and whether we could trust the empirically observed success. As we show in Example 1 on page 5, as well as in the full proof of Theorem 5 (appendix), when the problem is unidentifiable, the learning algorithm might become confused about which noise rates are the correct ones. We have discussed in the opening paragraph of the introduction that the knowledge of noise rate often plays an important role in the problem of learning with noisy label.
>
> In addition to these toy examples, the literature has observed evidence that when the learning with noisy label problem becomes not identifiable, the chance of estimating the transition matrix correctly will be even lower. For instance, in [1], the authors show that when the quality of representations dropped, the estimation error in the noise transition matrix increases significantly (Figure 1 therein). Other previous references have documented these challenges too [2].
>
> We now have better motivated the study of identifiability in the introduction.
>
> **Why instance-dependent label noise?**
>
> Instance-dependent label noise model/setting has gained popularity recently. Before this surge, the typical assumption made in the literature is that the label noise is generated according to a class-dependent model, but independent from where the instance lies. One recent paper [4] has empirically shown that the above class-dependent model is not precise in capturing the real-world noise patterns, but rather real human-level noise patterns follow an instance-dependent setting. Intuitively, instance $X$ encodes the difficulties in generating the label for it. The recent literature on learning with noisy labels has also observed strong interest in exploring this more realistic, flexible, and powerful noise model to capture complicated noise patterns [5 - 7]. Therefore, it is important to understand whether a particular instance-dependent label noise is indeed identifiable and learnable or not.
>
> Despite the recent surging and empirical successes in handling instance-dependent label noise, it remained unclear whether the underlying learning problem is indeed encoding an identifiable noise model and in fact learnable. Before we can answer this question, the observed empirical successes will remain suspicious as it is unclear whether the observed performance improvements come from specific assumptions that restricted the noise model, or whether the observed improvements should be attributed to other factors (including even randomness in the experiments)
>
>
> **Experiments**
>
> We conduct an experiment on CIFAR10 and test the performance of forward loss correction [3] with different noise transition matrices. The experiment shows that when the used transition matrix is far from the ground truth, the performance is even worse than the baseline (vanilla training with cross-entropy loss). See detailed experimental setting and results in C.4.1 of supplementary material.
>
> In our controlled synthetic experiments, we also show a case where there is only one dimension of feature and one noisy label, the problem is hardly identifiable: in this case, calling our Theorem 7, we require the number of disentangled features to be $\geq 2$ but we only have 1. Indeed we observe a high error in estimating the noise transition matrix. in this case, forward loss correction [3] and cross-entropy loss nearly have the same performance.  See detailed experimental setting and results in C.4.2 of supplementary material.
>
> **Citation:** We fixed the citation style. Thank you for the catch.
>
> **Noise model**
>
> While Section 4 showed both the sufficiency and the necessity of three labels, Section 5 has generalized the discussions to broader settings that do not necessarily need 3 labels but rather each instance is only associated with one label.
>
> As discussed earlier, the instance-dependent noise model is the most flexible model to capture the dependency between noise rate and each instance [4]. Independently generated labels are a pretty common modeling assumption adopted both in the machine learning and crowdsourcing community. Typically in a label collection system, when randomizing the tasks sufficiently among independent workers, the labels are generated in a rather independent fashion [8].

---

> > ### Author Response · Authors · 2022-11-13
> > **Response to Reviewer G5Yr Part II (references)**
> >
> > References we used in the response:
> >
> > [1] Beyond Images: Label Noise Transition Matrix Estimation for Tasks with Lower-Quality Features. ICML 2022.
> >
> > [2] Are anchor points really indispensable in label-noise learning? NeurIPS 2020.
> >
> > [3] Making deep neural networks robust to label noise: A loss correction approach CVPR 2017
> >
> > [4] Learning with noisy labels revisited: A study using real-world human annotations. ICLR 2022
> >
> > [5] Learning with bounded instance-and label-dependent label noise. ICML 2020.
> >
> > [6] Learning with instance-dependent label noise: A sample sieve approach. ICLR 2021
> >
> > [7] Learning with feature-dependent label noise: A progressive approach. ICLR 2021
> >
> > [8] Leveraging Peer Communication to Enhance Crowdsourcing. WWW 2019.

---

> > > ### Comment · Reviewer_G5Yr · 2022-11-24
> > > **New comments**
> > >
> > > Thanks for your careful response. Most of my concerns have been addressed, but I still feel confused about the following points. Firstly, you mentioned that "noise rate often plays an important role in the problem of learning with noisy label", but the noise rate is unknown in real datasets. How can you evaluate the importance of the noise rate in real data? Additionally, you claimed that "the instance-dependent noise model is the most flexible model" in the field of learning with noisy label, this may be too absolute. Personally I think that the label noise is not only instance-dependent, but also may contain some relationships between different instances, which seems to be more flexible.

---

> > > > ### Author Response · Authors · 2022-11-24
> > > > **follow ups**
> > > >
> > > > Both are great questions. For the first one on evaluating noise rate in real data, so far the community has been curating benchmark datasets that contain both ground truth labels (for verification purpose, collected with more efforts and control) and the noisy labels. See for example CIFAR-10H:
> > > >
> > > > https://github.com/jcpeterson/cifar-10h
> > > >
> > > > and CIFAR-N:
> > > >
> > > > http://noisylabels.com/
> > > >
> > > > Using this data, the research results can be validated.
> > > >
> > > > In a real-world application, the typical setting assumed is also that though training data is collected with noise, the effectiveness of the model will be tested against the real-world scenario. For example, a facial recognition tool might be trained using noisy training data, but its real performance in the field is measured by the true data patterns.
> > > >
> > > > Regarding the second question, this is a great comment. Indeed, the noisy label community has been mostly focused on settings where the noisy labels for each instance is generated independently. Within this context, instance-dependent model gives us the freedom to encode the dependence between the noise rate and each instance $X$. Through $T(X)$, we can capture the relationship between multiple instances, for example, that similar instances $X_1$ and $X_2$ might enjoy similar noise rates $T(X_1)$ and $T(X_2)$.
> > > >
> > > > But the reviewer is correct that more complicated noise model exists: if we understand correctly, one possibility is that what if the generation of noisy labels (not only the noise rates) for multiple instances are correlating with each other. To our best knowledge, this is a very much under-studied topic and would certainly be of future interests to the research community. We will clarify this.

---

> > > > > ### Author Response · Authors · 2022-12-03
> > > > > **gentle follow up**
> > > > >
> > > > > Dear Reviewer G5Yr,
> > > > >
> > > > > Thank you for your comments and discussions. We are following up to see if there is anything that we could further clarify. Thank you.
> > > > >
> > > > > Best,
> > > > >
> > > > > Authors

---

### Author Response · Authors · 2022-11-13
**Revision uploaded**

Dear reviewers,

We have uploaded our revision based on your constructive comments. We are grateful for many of them, which we believe have helped better motivate and position this current draft. In particular,

-  we better motivated this study in the introduction and have added supporting experiments in Appendix C.4 (wrong noise transition matrix can be harmful & what happens when the problem is not guaranteed to be identifiable)
- we better explained Theorem 5 and generalized the argument for the multi-class setting.
- we added explanations for the implication of Theorem 6 (Appendix page 19) and disentangled features (page 8)
- we followed most of the other suggestions and have revised the paper accordingly.

We have highlighted the major changes in color blue.

Thank you! and we look forward to continuing the discussion.

Best,

Authors

---

### Decision · Program_Chairs · 2023-01-20

**Decision:**

Reject

**Justification For Why Not Higher Score:**

Thanks for the responses to the reviewers' questions, which clarified some of the concerns they had in their initial reviews. This paper investigates the important problem of identifying the noise transition matrix as a function of instances. The obtained condition through Kruskal's results is interesting. Overall, I found the result of this paper worth publishing, but through the thorough discussions between the authors and reviewers, I must say that the current manuscript requires non-trivial major revision to clarify the presentation of the results. Therefore, I cannot recommend the current manuscript for acceptance.

**Justification For Why Not Lower Score:**

N/A

**Metareview: Summary, Strengths And Weaknesses:**

Summary:
This paper shows the identifiability condition of the noise transition matrix in the instance-dependent noise setup.

Strength:
Using Kruskal's results to investigate the identifiability of label noise is novel and interesting. The result that disentangled features help  identify the noise transition matrix in the instance-dependent noise setup is valuable.

Weakness:
The clarity of the current manuscript is low and the reviewers were confused with the notation. Also, the justification of the 2-NN idea still requires more discussion to be broadly acceptable to the readers.